# Better LMO-based Momentum Methods with Second-Order Information

## Abstract

The use of momentum in stochastic optimization algorithms has shown empirical success across a range of machine learning tasks. Recently, a new class of momentum-based stochastic algorithms has emerged within the Linear Minimization Oracle (LMO) framework–leading to methods such as Muon, Scion, and Gluon–for effectively solving deep neural network training problems. However, traditional stochastic momentum methods offer convergence guarantees no better than $\mathcal{O}(1/K^{1/4})$. While several approaches–such as Hessian-Corrected Momentum (HCM)–have aimed to improve this rate, their theoretical results are generally restricted to the Euclidean norm setting. This limitation hinders their applicability in the problems where arbitrary norms are often required. In this paper, we extend the LMO-based framework by integrating HCM, and provide the convergence guarantees under relaxed smoothness and arbitrary norm settings. Specifically, we establish improved convergence rates of $\mathcal{O}(1/K^{1/3})$ for HCM, thereby surpassing the classical momentum rate and allowing the algorithms to better adapt to the geometry of the problem. Experimental results on training Multi-Layer Perceptrons (MLPs) and Long Short-Term Memory (LSTM) networks support our theoretical findings, demonstrating that the proposed LMO-based algorithms with HCM significantly outperform their vanilla algorithms with traditional momentum.

## 1 Introduction

Stochastic momentum methods have been widely used for solving data-intensive machine learning problems. They are employed as the default Stochastic Gradient Descent (SGD) optimizer in open-sourced software libraries, such as Pytorch (Paszke et al., 2019), for deep neural network (DNN) training. The concept of stochastic momentum methods is inspired by deterministic momentum methods, including Polyak's heavy-ball method (Polyak, 1964). The heavy-ball method achieves an accelerated linear convergence rate compared to classical gradient descent (Ghadimi et al., 2015) for minimizing twice continuously differentiable strongly convex and smooth functions.

The convergence for stochastic momentum methods has been well studied for minimizing an $L$-smooth function $f(x)$, i.e. the function whose gradient is $L$-Lipschitz continuous. Most prior works have focused on the Euclidean norm setting, including convergence analyses by Yan et al. (2018); Yu et al. (2019); Liu et al. (2020); Cutkosky & Mehta (2020); Hübler et al. (2024a). More recently, there has been increasing interest in extending these results to arbitrary norm settings, as demonstrated by Kovalev (2025); Pethick et al. (2025a); Riabinin et al. (2025). Despite their empirical success over SGD, no existing theoretical analysis has established a provable advantage for momentum. In particular, their convergence guarantees do not improve upon the $\mathcal{O}\left(1/K^{1/4}\right)$ convergence in the expected gradient norm.

However, many learning applications, including distributionally robust optimization (Chen et al., 2023) and deep neural network training, often violate the standard $L$-smoothness condition. For instance, many empirical studies have shown that the Hesian norm is not uniformly upper-bounded as required by $L$-smoothnes when training neural networks like LSTM by Zhang et al. (2019), ResNet20 by Zhang et al. (2019), and transformers by Crawshaw et al. (2022). These observations have motivated the development of relaxed smoothness conditions that more accurately reflect the behavior of modern learning problems. Examples of relaxed smoothness conditions include the $(L_0, L_1)$-smoothness conditions (Zhang et al., 2019; Gorbunov et al., 2024; Chen et al.,

2023; Vankov et al., 2024), the $(L_0, L_1, \alpha)$-smoothness conditions (Chen et al., 2023), and the $\ell$-smoothness conditions (Li et al., 2023).

Under relaxed smoothness conditions, several first-order methods have been studied. Under $(L_0, L_1)$-smoothness, the $\mathcal{O}(1/K^{1/4})$ convergence in the expected gradient norm has been established for SGD (Li et al., 2023), clipped SGD (Zhang et al., 2019; 2020a; Koloskova et al., 2023), and AdaGrad-Norm (Faw et al., 2023; Wang et al., 2023). Also, the same $\mathcal{O}(1/K^{1/4})$ rate under $(L_0, L_1)$-smoothness has been proved for stochastic momentum methods, in the Euclidean norm setting (Hübler et al., 2024b; Zhao et al., 2021) and the arbitrary norm setting (Riabinin et al., 2025).

To improve the convergence of stochastic momentum methods, several approaches have been recently proposed to modify gradient estimators in momentum updates. One such approach, called extrapolated momentum, was introduced by Cutkosky & Mehta (2021), and achieves a convergence rate of $\mathcal{O}(1/K^{2/5})$. This convergence was further improved by recent momentum variants, such as STORM (Cutkosky & Orabona, 2019), MARS (Yuan et al., 2024), and second-order momentum (Salehkaleybar et al., 2022; Tran & Cutkosky, 2022; Zhang et al., 2020b). These momentum variants ensure the improved rate of $\mathcal{O}(1/K^{1/3})$, proven to be optimal for minimizing smooth nonconvex functions under mild conditions (Arjevani et al., 2023). However, these methods have been primarily analyzed under traditional smoothness assumptions and within the Euclidean setting, limiting their applicability in broader machine learning contexts, such as neural network training.

## 2 CONTRIBUTIONS

The goal of this paper is to generalize two variants of second-order momentum methods (Salehkaleybar et al., 2022; Tran & Cutkosky, 2022) for minimizing nonconvex functions under relaxed smoothness assumptions in the arbitrary norm setting. We make the following contributions:

First, we incorporate two second-order momentum update variants into the class of optimization algorithms that leverage a Linear Minimization Oracle (LMO) proposed by Pethick et al. (2025a); Kovalev (2025). Our methods generalize the second-order momentum techniques originally proposed in the Euclidean setting by Salehkaleybar et al. (2022); Tran & Cutkosky (2022) to broader geometries and norm choices corresponding to the problems.

Second, we establish an $\mathcal{O}(1/K^{1/3})$ convergence in the expected gradient norm for LMO-based methods using two second-order momentum variants under relaxed smoothness assumptions in the arbitrary norm setting. This improves upon the previously known $\mathcal{O}(1/K^{1/4})$ rate for LMO-based methods using Polyak momentum by Pethick et al. (2025a;b); Riabinin et al. (2025); Kovalev (2025). Also, our results match the known rates for second-order momentum methods under standard $L$-smoothness in the Euclidean setting (Salehkaleybar et al., 2022; Tran & Cutkosky, 2022). A comparison of theoretical guarantees is summarized in Table 1.

Third, we empirically evaluate LMO-based methods with second-order momentum and other momentum updates on three nonconvex problems: logistic regression problems, Multi-Layer Perceptron (MLP) training problems, and Long Short-Term Memory (LSTM) network training problems. Our results validate our theory, illustrating that LMO-based methods using second-order momentum outperform those using Polyak momentum and extrapolated momentum.

## 3 RELATED WORKS

**Stochastic momentum methods.** Momentum is widely used to expedite the training process in stochastic optimization methods. In the Euclidean norm setting, stochastic momentum methods enjoy the $\mathcal{O}(1/K^{1/4})$ convergence under traditional smoothness (Yan et al., 2018; Yu et al., 2019; Liu et al., 2020; Cutkosky & Mehta, 2020), and recently under relaxed smoothness (Hübler et al., 2024a). Moreover, momentum has been incorporated to enhance the performance of other algorithms, such as distributed methods with communication compression (Fatkhullin et al., 2023; Khirirat et al., 2024), and LMO-based optimization methods for deep neural network training (Jordan et al., 2024; Pethick et al., 2025a). To improve upon the $\mathcal{O}(1/K^{1/4})$ convergence rate, several novel momentum updates have been proposed. For instance, stochastic methods employing extrapolated momentum (Cutkosky & Mehta, 2021) have been shown to achieve an improved rate of $\mathcal{O}(1/K^{2/7})$. Fur-

| Methods | Rate | Gradient Smoothness | Hessian Smoothness | Norm Type |
|---|---|---|---|---|
| Unconstrained SCG Pethick et al. (2025a) | $\mathcal{O}\left(1/K^{1/4}\right)$ | $(L)$ | – | Arbitrary |
| GGNC Pethick et al. (2025b) | $\mathcal{O}\left(1/K^{1/4}\right)$ | $(L_0, L_1)$ | – | Arbitrary |
| Gluon Riabinin et al. (2025)[Theorem 2] | $\mathcal{O}\left(1/K^{1/4}\right)$ | $(L_0, L_1)$ | – | Arbitrary |
| Second-order momentum (Variant 1) Salehkaleybar et al. (2022) | $\mathcal{O}\left(1/K^{1/3}\right)$ | $L$ | – | Euclidean |
| Second-order momentum (Variant 2) Tran & Cutkosky (2022) | $\mathcal{O}\left(1/K^{1/3}\right)$ | $L$ | $M$ | Euclidean |
| Second-order momentum (Variant 1) **NEW** (Theorem 1 ) | $\mathcal{O}\left(1/K^{1/3}\right)$ | $(L_0, L_1)$ | – | Arbitrary |
| Second-order momentum (Variant 2) **NEW** (Theorem 2) | $\mathcal{O}\left(1/K^{1/3}\right)$ | $(L_0, L_1)$ | $(M_0, M_1)$ | Arbitrary |

Table 1: Comparisons of convergence for LMO-based methods that use Polyak momentum like Gluon, and LMO-based methods that use second-order momentum methods to attain
$$\min_{k=\{0,1,\dots,K\}} \mathrm{E}\left[\left\|\nabla f(x^k)\right\|_\star\right] \leq \varepsilon.$$

thermore, the methods based on STORM (Cutkosky & Orabona, 2019), MARS (Yuan et al., 2024), and Hessian-corrected momentum techniques (Salehkaleybar et al., 2022; Tran & Cutkosky, 2022; Zhang et al., 2020a) attain the convergence rate of $\mathcal{O}(1/K^{1/3})$, which matches the known lower bound established by Arjevani et al. (2023). However, the convergence guarantees of these novel stochastic momentum methods were restricted to traditional smoothness in the Euclidean setting. In this paper, we will incorporate Hessian-corrected momentum techniques into the LMO-based optimization methods to guarantee the $\mathcal{O}(1/K^{1/3})$ convergence in the arbitrary norm setting, which improves upon the $\mathcal{O}(1/K^{1/4})$ convergence achieved by current Muon optimizers (i.e., LMO-based methods using Polyak momentum).

**Muon.** Muon optimizers (Jordan et al., 2024) update the model parameters (with their inherent matrix structure) with orthogonalized gradient momentum. Empirical studies, e.g. by Jordan et al. (2024); Liu et al. (2025a), highlight the superior performance of Muon over Shampoo (Gupta et al., 2018), SOAP (Vyas et al., 2024), and AdamW (Loshchilov & Hutter, 2017), for language model training. These encouraging observations motivate the studies of the convergence analysis for Muon. Originally, the convergence of Muon was analyzed by Li & Hong (2025) under the $L$-smoothness with respect to the Frobenius norm. Later, LMO-based optimization methods with momentum were proposed by Pethick et al. (2025a) to capture optimizers, including Muon, Scion, and stochastic Euclidean momentum methods with momentum. The methods were shown to enjoy the $\mathcal{O}(1/K^{1/4})$ convergence under traditional smoothness (Pethick et al., 2025a; Kovalev, 2025) and under relaxed smoothness (Pethick et al., 2025b; Riabinin et al., 2025), in the arbitrary norm setting.

**Relaxed smoothness.** Training deep neural networks, especially large-scale language models, poses significant challenges due to their highly non-convex functions. To better capture this, a variety of relaxed smoothness conditions on the functions have been proposed. One early contribution by Zhang et al. (2019) introduced a function class in which the Hessian norm grows linearly with the gradient norm, providing a more flexible alternative to classical smoothness assumptions. Building upon this, several works have introduced broader classes of functions, such as the symmetric $(L_0, L_1, \alpha)$-smoothness condition (Chen et al., 2023), the $\ell$-smoothness condition (Li et al., 2023), and the $(\rho, K_0, K_\rho)$-smoothness condition (Liu et al., 2025b). These conditions aim to more accurately reflect the optimization landscape encountered in deep learning. Under such relaxed assumptions, the convergence behavior of various gradient-based methods has been extensively studied (Crawshaw et al., 2022; Hübler et al., 2024a; Zhang et al., 2019; Koloskova et al., 2023; Chezhegov et al., 2025; Gorbunov et al., 2024; Vankov et al., 2024; Khirirat et al., 2024), providing theoretical insights that align more closely with empirical observations in neural network training.

## 4 PROBLEM FORMULATION

We consider the following unconstrained stochastic optimization problems

$$\min_{x \in \mathbb{R}^d} f(x) := \mathrm{E}_{\xi \sim \mathcal{D}}\left[f_\xi(x)\right], \tag{1}$$

where $f_\xi(x)$ is a differentiable but possibly nonconvex function, $x \in \mathbb{R}^d$ is the $d$-dimensional vector of model parameters, and $\xi$ is a random vector, which represents a training data sample drawn from an unknown data distribution $\mathcal{D}$.

Often, we access a stochastic oracle to compute the stochastic gradient and Hessian of $f$ at $x$ with respect to $\xi \sim \mathcal{D}$ denoted by $\nabla f_\xi(x)$ and $\nabla^2 f_\xi(x)$, respectively. We assume that the stochastic gradient and stochastic Hessian satisfy the following unbiased and variance-bounded properties:

**Assumption 1.** $\nabla f_\xi(x)$ and $\nabla^2 f_\xi(x)$ is an unbiased and variance-bounded estimator of $\nabla f(x)$ and $\nabla^2 f(x)$, respectively, i.e. for all $x, w \in \mathbb{R}^d$,

$$\mathrm{E}_\xi\left[\nabla f_\xi(x)\right] = \nabla f(x), \quad \mathrm{E}_\xi\left[\left\|\nabla f_\xi(x) - \nabla f(x)\right\|_2^2\right] \le \sigma_g^2,$$

$$\mathrm{E}_\xi\left[\nabla^2 f_\xi(x)\right] = \nabla^2 f(x), \quad \text{and} \quad \mathrm{E}_\xi\left[\left\|(\nabla^2 f_\xi(x) - \nabla^2 f(x))w\right\|_2^2\right] \le \sigma_H^2 \left\|w\right\|_2^2.$$

Assumption 1 has been commonly used for analyzing the convergence of optimization algorithms using stochastic gradients (Ghadimi & Lan, 2013; Cutkosky & Orabona, 2019; Gorbunov et al., 2020) and stochastic Hessians (Tran & Cutkosky, 2022; Cutkosky & Orabona, 2019; Jiang et al., 2024).

To facilitate our analysis, we also impose the following standard assumptions on objective functions.

**Assumption 2.** The function $f : \mathbb{R}^d \to \mathbb{R}$ is bounded from below, i.e., $f^{\inf} = \inf_{x \in \mathbb{R}^d} f(x) > -\infty$.

**Assumption 3.** The function $f : \mathbb{R}^d \to \mathbb{R}$ is, and its gradient $\nabla f(x)$ is symmetrically $(L_0, L_1)$-Lipschitz continuous with respect to the norm $\|\cdot\|$, i.e. for all $x, y \in \mathbb{R}^d$,

$$\|\nabla f(x) - \nabla f(y)\|_\star \le \left(L_0 + L_1 \sup_{\theta \in [0,1]} \|\nabla f(\theta x + (1-\theta)y)\|_\star\right) \|x - y\|.$$

**Assumption 4.** The function $f : \mathbb{R}^d \to \mathbb{R}$ is twice differentiable, and its Hessian $\nabla^2 f(x)$ is symmetrically $(M_0, M_1)$-Lipschitz continuous with respect to the norm $\|\cdot\|$, i.e. for all $x, y \in \mathbb{R}^d$,

$$\left\|\nabla^2 f(x) - \nabla^2 f(y)\right\|_\star \le \left(M_0 + M_1 \sup_{\theta \in [0,1]} \|\nabla f(\theta x + (1-\theta)y)\|_\star\right) \|x - y\|.$$

Assumptions 2, 3, and 4 ensure the lower-bound of the objective function $f$, the symmetric relaxed Lipschitz continuity of its gradient $\nabla f$, and the symmetric relaxed Lipschitz continuity of its Hessian $\nabla^2 f$, respectively. On the one hand, Assumption 3 is a generalization of the symmetric $(L_0, L_1)$-Lipschitz continuity of $\nabla f(\cdot)$ with respect to the Euclidean norm by Gorbunov et al. (2024); Chen et al. (2023). Also, Assumption 3 recovers $L$-Lipschitz continuity with respect to the arbitrary norm by Kovalev (2025); Pethick et al. (2025a) when we let $L_0 = L$ and $L_1 = 0$. On the other hand, Assumption 4 is a generalization of the asymmetric $(M_0, M_1)$-Lipschitz continuity of the Hessian with respect to the Euclidean norm by Xie et al. (2024, Assumption 3). Also, Assumption 4 obtains the $L_H$-Lipschitz continuity of the Hessian by Carmon et al. (2018); Nesterov & Polyak (2006), when we let $M_0 = L_H$ and $M_1 = 0$.

Finally, note that the variance (Assumption 1) is measured with respect to the Euclidean norm, while the Lipschitz continuity of the gradient (Assumption 3) and Hessian (Assumption 4) with respect to the arbitrary norm and its dual norm. Hence, we describe a connection between these norms using the following inequality:

$$\underline{\rho} \left\|x\right\|_2 \le \left\|x\right\|_\star \le \bar{\rho} \left\|x\right\|_2, \quad \text{and} \quad \underline{\theta} \left\|x\right\|_2 \le \left\|x\right\| \le \bar{\theta} \left\|x\right\|_2. \tag{2}$$

### 4.1 LMO-BASED METHODS WITH MOMENTUM

To solve the problem in equation 1 for neural network training tasks, we focus on the LMO-based optimization methods using momentum (Pethick et al., 2025a; Kovalev, 2025), which update the iterates $\{x_k\}_{k \ge 0}$ as follows: Given $x_0, m_0 \in \mathbb{R}^d$,

$$x_{k+1} = x_k + \mathrm{lmo}(m_k), \quad \text{and} \quad m_{k+1} = (1 - \alpha_k)m_k + \alpha_k \nabla f_{\xi_{k+1}}(x_{k+1}). \tag{3}$$

Here, $\eta_k > 0$ is a stepsize, $\alpha_k \in (0, 1)$ is a momentum parameter, $\nabla f_\xi(x)$ is the stochastic gradient, and $\mathrm{lmo}(m_k) := \mathrm{argmin}_{\|x\| \leq \eta_k} \langle m_k, x \rangle$. The methods in equation 3 encompass a class of stochastic momentum methods under specific norm choices. For instance, they recover normalized stochastic momentum methods (Cutkosky & Mehta, 2020) when we let $\|\cdot\| = \|\cdot\|_2$, and sign stochastic momentum methods (Jiang et al., 2025) when we let $\|\cdot\| = \|\cdot\|_\infty$.

The LMO-based methods using momentum in equation 3 were theoretically shown by Pethick et al. (2025a); Kovalev (2025); Riabinin et al. (2025) to attain the $\mathcal{O}\left(1/K^{1/4}\right)$ convergence rate in the gradient norm in the arbitrary norm setting, which recovers the same rate as stochastic momentum methods in the Euclidean norm setting, e.g. by Cutkosky & Mehta (2020); Jiang et al. (2025); Zhang et al. (2020a); Hübler et al. (2024b).

To further improve the iteration complexity of the methods in equation 3, many momentum variants have been proposed.

**Extrapolated momentum.** The first variant is extrapolated momentum (Cutkosky & Mehta, 2021), which updates the momentum vector $m_k$ according to

$$m_{k+1} = (1 - \alpha_k)m_k + \alpha_k \nabla f_{\xi_{k+1}}(y_{k+1}), \tag{4}$$

where $y_{k+1} = x_{k+1} + (^{(1-\alpha_k)}/\alpha_k)(x_{k+1} - x_k)$. LMO-based methods using extrapolated momentum enjoys the $\mathcal{O}(1/K^{2/7})$ convergence rate under traditional smoothness in the arbitrary norm setting Kovalev (2025), which matches the rate in the Euclidean norm setting by Cutkosky & Mehta (2021, Theorem 5) with $p = 2$.

**STORM.** To further improve the convergence of stochastic methods using extrapolated momentum, we may consider Stochastic Recursive Momentum (STORM) (Cutkosky & Orabona, 2019). The key technique is to use the stochastic gradient difference $\nabla f_{\xi_{k+1}}(x_{k+1}) - \nabla f_{\xi_{k+1}}(x_k)$ as the correction term to expedite the training process. In STORM, the momentum vector $m_k$ is updated via

$$m_{k+1} = (1 - \alpha_k)(m_k + [\nabla f_{\xi_{k+1}}(x_{k+1}) - \nabla f_{\xi_{k+1}}(x_k)]) + \alpha_k \nabla f_{\xi_{k+1}}(x_k). \tag{5}$$

LMO-based methods using STORM have been recently shown to achieve the $\mathcal{O}(1/K^{1/3})$ convergence rate under relaxed smoothness in the Frobenius norm setting recently by Huang et al. (2025), which recovers the rate under traditional smoothness in the Euclidean norm setting by Cutkosky & Orabona (2019). To better control the gradient correction term in STORM, Yuan et al. (2024) introduce a scaling parameter $\beta_k \in (0, 1)$. The resulting algorithm, named MARS, modifies STORM by adjusting the update rule for the momentum vector $m_k$ as follows:

$$m_{k+1} = (1 - \alpha_k)\left(m_k + \frac{\beta_k}{1 - \alpha_k}[\nabla f_{\xi_{k+1}}(x_{k+1}) - \nabla f_{\xi_{k+1}}(x_k)]\right) + \alpha_k \nabla f_{\xi_{k+1}}(x_k). \tag{6}$$

By tuning the scaling factor $\beta_k$ properly, MARS can provide the finer control over the gradient difference term, thus leading to improved convergence performace.

**Second-order momentum.** For achieving the same convergence in stochastic optimization, second-order momentum (Salehkaleybar et al., 2022; Tran & Cutkosky, 2022; Zhang et al., 2020b), called Hessian-corrected momentum, can serve as an alternative to STORM. It utilizes the Hessian-vector product as the correction term to improve the complexity of momentum-based methods. It performs the following update:

$$m_{k+1} = (1 - \alpha_k)(m_k + \nabla^2 f_{\xi_{k+1}}(\hat{x}_{k+1})(x_{k+1} - x_k)) + \alpha_k \nabla f_{\xi_{k+1}}(x_{k+1}). \tag{7}$$

The second-order momentum update in equation 7 becomes the one proposed by Tran & Cutkosky (2022) when $\hat{x}_{k+1} = x_{k+1}$, and by Salehkaleybar et al. (2022) when $\hat{x}_{k+1} = b_k x_{k+1} + (1 - b_k)x_k$ with $b_k \in \mathbb{R}$ generated by the uniform distribution $\mathcal{U}(0, 1)$. Like STORM, second-order momentum ensures the $\mathcal{O}(1/K^{1/3})$ convergence, which is faster than the $\mathcal{O}(1/K^{2/7})$ convergence of extrapolated momentum and the $\mathcal{O}(1/K^{1/4})$ convergence of Polyak momentum. However, the results of second-order momentum are limited to traditional smoothness in the Euclidean norm setting, which reduces its applicability, especially in solving neural network training problems.

## 5 LMO-BASED METHODS WITH SECOND-ORDER MOMENTUM

Now, we study LMO-based methods (Pethick et al., 2025a; Kovalev, 2025) using two second-order momentum variants (Salehkaleybar et al., 2022; Tran & Cutkosky, 2022; Zhang et al., 2020b). The methods leverage the LMO-based oracle in equation 3, second-order momentum updates in equation 7, and the scaling factor $\beta_k$ introduced in MARS (Yuan et al., 2024). See Algorithm 1 for the detailed description of the methods.

In Algorithm 1, the second-order momentum incorporates the Hessian-vector product term into its update to accelerate the training process using the LMO-based momentum methods. Computing the Hessian-vector product can take roughly the same time as computing the gradient (Pearlmutter, 1994). This can be achieved by using the automatic differentiation package to compute $\nabla^2 f(x)v$ for $x, v \in \mathbb{R}^d$ by evaluating $\nabla h(x)$, where $h(x) = \langle \nabla f(x), v \rangle$ (Tran & Cutkosky, 2022).

---

**Algorithm 1** LMO-based Optimization Methods with Second-order Momentum

1: **Input:** Tuning parameters $\eta_k > 0$ and $\alpha_k, \beta_k \in (0, 1)$ for $k = 0, 1, \ldots$; Initial points $x_0, m_0 \in \mathbb{R}^d$; Total number of iterations $K > 0$
2: **for** each iteration $k = 0, 1, \ldots, K$ **do**
3:      Update $x_{k+1} = x_k + \text{lmo}(m_k)$, where $\text{lmo}(m_k) := \underset{\|x\| \leq \eta_k}{\arg\min} \langle m_k, x \rangle$
4:      Sample $\xi_{k+1} \sim \mathcal{D}$
5:      **if** Variant 1 **then**
6:         Sample $b_{k+1} \sim \mathcal{U}(0, 1)$
7:         Set $\hat{x}_{k+1} = b_{k+1} x_{k+1} + (1 - b_{k+1}) x_k$
8:         Update $m_{k+1} = (1 - \alpha_k) \left( m_k + \frac{\beta_k}{1 - \alpha_k} \nabla^2 f_{\xi_{k+1}}(\hat{x}_{k+1})(x_{k+1} - x_k) \right) + \alpha_k \nabla f_{\xi_{k+1}}(x_{k+1})$
9:      **else if** Variant 2 **then**
10:         Update $m_{k+1} = (1 - \alpha_k) \left( m_k + \frac{\beta_k}{1 - \alpha_k} \nabla^2 f_{\xi_{k+1}}(x_{k+1})(x_{k+1} - x_k) \right) + \alpha_k \nabla f_{\xi_{k+1}}(x_{k+1})$
11:      **end if**
12: **end for**
13: **Output:** $x^{K+1}$

---

## 6 CONVERGENCE THEOREMS

We present the convergence results for LMO-based methods using two variants of second-order momentum updates. Our results show that these methods achieve the $\mathcal{O}(1/K^{1/3})$ convergence under relaxed smoothness in the arbitrary norm setting.

**Theorem 1** (Variant 1 of Algorithm 1 )**.** Consider the problem of minimizing $f(x) = \mathbb{E}_{\xi \sim \mathcal{D}} [f_\xi(x)]$. Let $f$ be twice differentiable, and let Assumption 1, 2, and 3 hold. Then, the iterates $\{x_k\}$ generated by Variant 1 of Algorithm 1 with $\beta_k = 1 - \alpha_k$ and with

$$\alpha_k = \alpha = \frac{1}{(K+1)^{2/3}}, \quad \text{and} \quad \eta_k = \eta = \frac{\hat{\eta}}{(K+1)^{2/3}},$$

where $\hat{\eta} = \frac{1}{80 L_1} \left( \frac{\bar{\rho}}{\underline{\rho}} \right)^{-1}$, satisfy

$$
\min_{k \in \{0, 1, \ldots, K\}} \mathbb{E}\left[ \|\nabla f(x_k)\|_\star \right] \leq \frac{2(f(x_0) - f_{\inf})}{\hat{\eta}(K+1)^{1/3}} + \frac{4 \|e_0\|_\star}{(K+1)^{1/3}} + \frac{4\bar{\rho}\hat{\eta}\hat{B}}{(K+1)^{1/3}}
$$

$$
+ 4\bar{\rho} \frac{1}{(K+1)^{1/3}} \sigma_g + L_0 \exp(L_1 \hat{\eta}) \frac{\hat{\eta}}{(K+1)^{2/3}},
$$

where $\hat{B} = \frac{3}{\underline{\rho}} L_0 (\exp(L_1 \hat{\eta}) + \sqrt{L_1 \hat{\eta}} \exp(L_1 \hat{\eta}) + 2) + \frac{2\sigma_H}{\underline{\theta}}$.

**Theorem 2** (Variant 2 of Algorithm 1). Consider the problem of minimizing $f(x) = \mathrm{E}_{\xi \sim \mathcal{D}}\left[f_\xi(x)\right]$. Let $f$ be twice differentiable, and let Assumptions 1, 2, 3, and 4 hold. Then, the iterates $\{x_k\}$ generated by Variant 2 of Algorithm 1 with $\beta_k = 1 - \alpha_k$ and with

$$\alpha_k = \alpha = \frac{1}{(K+1)^{2/3}}, \quad \text{and} \quad \eta_k = \eta = \frac{\hat{\eta}}{(K+1)^{2/3}},$$

where $\hat{\eta} = \frac{1}{3}\min\left\{\frac{1}{L_1}, \frac{1}{\sqrt{M_1}}\right\}$ satisfy

$$\min_{k \in \{0,1,\dots,K\}} \mathrm{E}\left[\|\nabla f(x_k)\|_\star\right] \leq \frac{2(f(x_0) - f_{\inf})}{\hat{\eta}(K+1)^{1/3}} + \frac{\hat{\eta}L_0 \exp(L_1\hat{\eta})}{(K+1)^{2/3}} + \frac{4\|e_0\|_\star}{(K+1)^{1/3}}$$

$$+ 2\left(M_0 + \frac{2M_1}{3}L_0\exp(L_1\hat{\eta})\right)\frac{\hat{\eta}^2}{(K+1)^{2/3}}$$

$$+ 4\frac{\bar{\rho}}{\underline{\theta}}\frac{\hat{\eta}\sigma_H}{(K+1)^{1/3}} + 4\frac{\bar{\rho}\sigma_g}{(K+1)^{1/3}}.$$

From Theorems 1 and 2, Algorithm 1 achieves the $\mathcal{O}(1/K^{1/3})$ convergence in the expected gradient norm for minimizing symmetric $(L_0, L_1)$-smooth functions with respect to the arbitrary norm. Our methods match the convergence rates of second-order momentum methods established under $L$-smoothness with respect to the Euclidean norm by Salehkaleybar et al. (2022); Tran & Cutkosky (2022), and do not require large minibatch sizes for computing stochastic gradients. Our results do not impose restrictive assumptions, such as bounded stochastic gradient norms by Tran & Cutkosky (2022). Moreover, our methods converge faster than LMO-based optimization methods using Polyak momentum analyzed by Pethick et al. (2025a;b); Kovalev (2025); Riabinin et al. (2025). Further note that, Variant 2 of Algorithm 1 assume the symmetric smoothness on the Hessian, which Variant 1 of Algorithm 1 does not. Moreover, although our theoretical results are established for Algorithm 1 with $\beta_k = 1 - \alpha_k$, the analysis can be extended to the case of general $\beta_k \in (0, 1]$. In such cases, the convergence bounds include an additional term depending on the choice of $\beta_k$.

We can leverage our analysis to establish the convergence of LMO-based methods in equation 3, which incorporate extrapolated momentum in equation 4, for minimizing relaxed smooth functions in the arbitrary norm setting (see Appendix E). The resulting convergence matches the $\mathcal{O}(1/K^{2/7})$ rate obtained by Kovalev (2025, Corollary 5) under traditional smoothness.

## 7 NUMERICAL EXPERIMENTS

We conduct numerical experiments to investigate the performance of LMO-based methods using Polyak momentum, extrapolated momentum (Cutkosky & Mehta, 2021), and two second-order momentum variants (Algorithm 1). Specifically, methods using two second-order momentum variants with $\beta_k = 1 - \alpha_k$ are referred to as SOM-V1 and SOM-V2, while those with general $\beta_k$ values are denoted by $\beta$-SOM-V1 and $\beta$-SOM-V2. We implemented these algorithms using PyTorch (Paszke et al., 2019), and benchmarked them for two nonconvex problems that satisfy symmetric $(L_0, L_1)$ smoothness: problems of training Multi-Layer Perceptrons (MLPs) and Long Short-Term Memory (LSTM) networks. We reported our results for logistic regression problems with nonconvex regulaization in Appendix F. For all experiments, each element of the initial point $x_0 \in \mathbb{R}^d$ was generated from the standard normal distribution $\mathcal{N}(0, 1)$, the random seed was fixed, and also the learning rate $\eta_k$ and the momentum parameter $\alpha_k$ were chosen as follows: (1) $\alpha_k = \frac{1}{\sqrt{k+1}}$ and $\eta_k = \frac{\eta_0}{(k+1)^{3/4}}$ for Polyak momentum, (2) $\alpha_k = \frac{1}{(k+1)^{4/7}}$ and $\eta_k = \frac{\eta_0}{(k+1)^{5/7}}$ for extrapolated momentum, and $\alpha_k = \frac{1}{(k+1)^{2/3}}$ and $\eta_k = \frac{\eta_0}{(k+1)^{2/3}}$ for all second-order momentum variants.

### 7.1 MLP

We evaluated the algorithms for binary classification tasks using the MLP model with two hidden layers over the `splice` dataset from the `libsvm` library (Chang & Lin, 2011). The dataset comprises 1000 training samples and 60 features. We minimize the objective function $f(x) = \mathcal{L}(x) + R(x)$, where $\mathcal{L}(x)$ is the binary cross-entropy loss with logits, and $R(x)$ is the nonconvex Welsch regularizer defined by $R(x) = \lambda \sum_{i=1}^d x_i^2/(1 + x_i^2)$ with $\lambda = 0.01$.

From Figure 1, LMO-based methods using second-order momentum outperform those using Polyak or extrapolated momentum. In particular, Variant 2 of second-order momentum achieves superior convergence in training loss, compared to other momentum variants. Furthermore, from Figure 2, the scaling factor $\beta_k$ further enhances the convergence achieved by second-order momentum. In particular, Variant 2 with the scaling factor $\beta_k$ exhibits the most consistent and strongest convergence performance in training loss, outperforming Variant 1 with the same scaling factor. Additionally, Figure 3 shows that increasing the mini-batch size for computing stochastic gradients improves the convergence performance of LMO-based methods using second-order momentum. Specifically, the methods using Variant 2 with the scaling factor $\beta_k$ achieve higher solution accuracy with a mini-batch size of 32 compared to sizes 1 and 16.

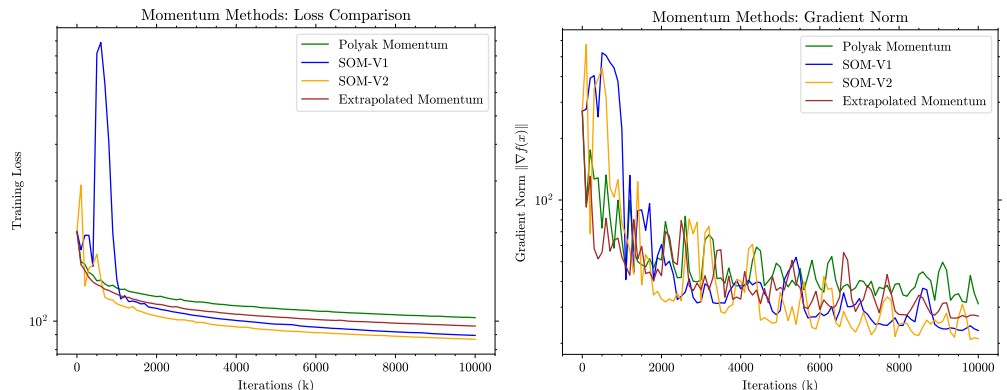

Figure 1: Comparison of training the MLP model on the `splice` dataset using LMO-based methods with various momentum updates in the training loss (left) and gradient norm (right).

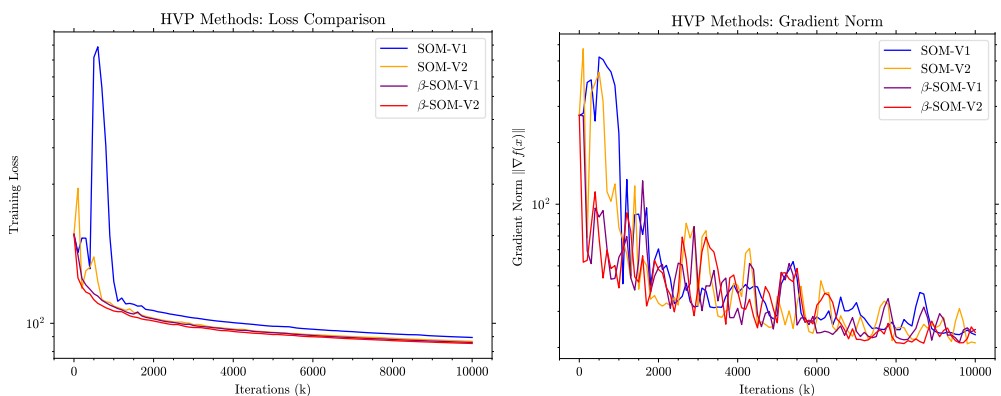

Figure 2: Comparison of training the MLP model on the `splice` dataset using LMO-based methods with second-order momentum (SOM) variants in the training loss (left) and gradient norm (right).

## 7.2 LSTM TRAINING

We benchmarked the algorithms for solving a word-level language modeling task using a two-layer LSTM network. The model consists of an embedding layer with 200 dimensions, followed by two stacked LSTM layers, each with 200 hidden units. A final linear decoder layer produces logits over the vocabulary for each input token. The experiments were conducted on the Penn Treebank (PTB) dataset, a standard dataset for evaluating language models. The dataset is split into training, validation, and test sets, with approximately $929,000$ tokens in the training set. The vocabulary is constructed from the unique words in the training data, with a total of $10,000$ tokens, including an end-of-sentence token `<eos>`. Tokenization is performed by splitting raw text on whitespace, converting each word into an index from the vocabulary. The model is trained using sequential

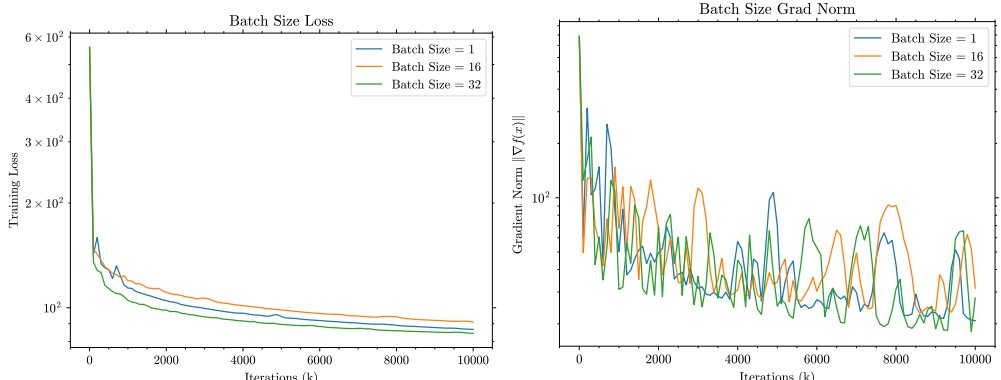

Figure 3: Performance of $\beta$-SOM-V2 (Variant 2 of second-order momentum with any $\beta_k$ scaling factor) when we vary the mini-batch size for training the MLP model on the `splice` dataset.

batches, employing Truncated Backpropagation Through Time (BPTT) with a sequence length of 35. For training, we used a minibatch size of 20 and the standard Cross-Entropy loss.

From Figure 4, we again observe the superior performance of LMO-based methods using Variant 2 of second-order mometnum and the scaling $\beta$ factor when we set the Euclidean norm (left) and the $\ell_\infty$-norm (right) in the LMO update.

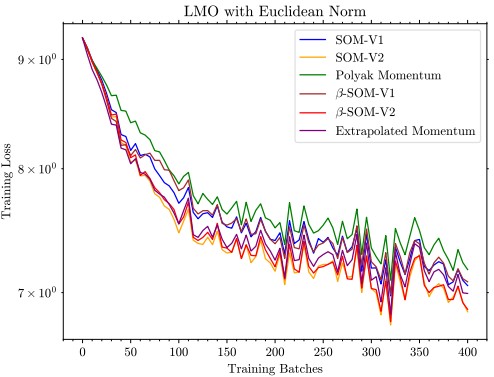

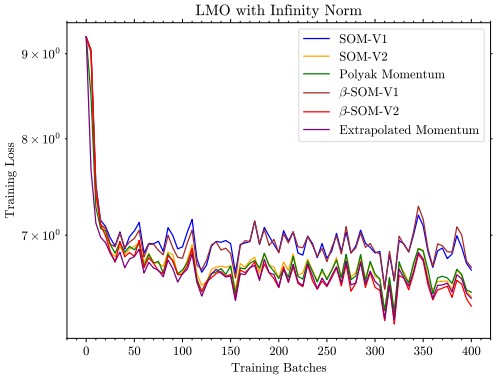

(a) Training loss vs. training batches of LMO-based methods with respect to the Euclidean norm $\|\cdot\|_2$. Other momentum variants attain superier performance to Polyak Momentum.

(b) Training loss vs. training batches of LMO-based methods with respect to the $\ell_\infty$-norm $\|\cdot\|_\infty$. Variant 2 of second-order momentum and extrapolated momentum outperforms Polyak momentum.

Figure 4: Comparison of LMO-based methods using momentum variants for the word-level language modeling task using the LSTM network on the Penn Treebank (PTB) dataset.

## 8 CONCLUSION

We have proposed LMO-based methods that incorporate two variants of second-order momentum for minimizing nonconvex functions under relaxed smoothness conditions in the arbitrary norm setting. The proposed methods achieve the convergence rate of $\mathcal{O}(1/K^{1/3})$ in the expected gradient norm, improving upon the $\mathcal{O}(1/K^{1/4})$ rate of existing LMO-based methods that use Polyak momentum. Notably, our theoretical results match the best-known rates for second-order momentum methods under standard smoothness assumptions in the Euclidean norm setting. Finally, our numerical experiments confirm the advantages of second-order momentum, consistently demonstrating superior convergence compared to Polyak and extrapolated momentum. Additionally, incorporating the adaptive scaling factor introduced in MARS, yields further improvements, thus enhancing both the convergence speed and solution accuracy of the proposed methods.

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

CONTENTS

## A  NOTATIONS

For random variables $u, v$, we use $\mathrm{E}\left[u\right]$ for the expectation of $u$, and $\mathrm{E}_v\left[u\right]$ for the expectation of $u$ with respect to $v$. For functions $f, g : \mathbb{R}^d \to \mathbb{R}$, we use $\nabla f$ and $\nabla^2 f$ to denote the gradient and Hessian of $f$, respectively. We write $f(x) = \mathcal{O}(g(x))$ to indicate that there exists a constant $C > 0$ and a value $x_0 \in \mathbb{R}$ such that $f(x) \leq C \cdot g(x)$ for all $x \geq x_0$. For vectors $x, y \in \mathbb{R}^d$, $\langle x, y \rangle = \sum_{i=1}^d x_i y_i$, while $\|x\|$, $\|x\|_\star$, $\|x\|_2$, and $\|x\|_\infty$ denote its arbitrary norm, associated dual norm, Euclidean norm, and $\ell_\infty$-norm, respectively.

## B  USEFUL LEMMA FROM RELAXED SMOOTHNESS

From Assumptions 3 and 4, we obtain the following lemma that is useful for our analysis.

**Lemma 1.** Let $f$ satisfy Assumptions 3 and 4. Then, for all $x, y \in \mathbb{R}^d$,

$$
\begin{aligned}
\|Z_f(x,y)\|_\star &\leq \frac{1}{2}(M_0 + M_1 \|\nabla f(y)\|_\star) \|x - y\|^2 \\
&\quad + \frac{1}{3}M_1(L_0 + L_1 \|\nabla f(y)\|_\star)\exp(L_1 \|x - y\|) \|x - y\|^3,
\end{aligned}
$$

where we denote $Z_f(x,y) := \nabla f(x) - \nabla f(y) - \nabla^2 f(y)(x - y)$.

*Proof.* Since $f$ is twice differentiable,

$$
\begin{aligned}
Z_f(x,y) &= \int_{\tau=0}^1 \nabla^2 f(x + \tau(y - x))(y - x)d\tau - \nabla^2 f(y)(x - y) \\
&= \int_{\tau=0}^1 [\nabla^2 f(x + \tau(y - x))(y - x) - \nabla^2 f(y)(x - y)]d\tau.
\end{aligned}
$$

Therefore,

$$
\|Z_f(x,y)\|_\star \leq \int_{\tau=0}^1 \left\|\nabla^2 f(x + \tau(y - x))(y - x) - \nabla^2 f(y)(x - y)\right\|_\star d\tau.
$$

Next, from Assumption 4,

$$
\begin{aligned}
\|Z_f(x,y)\|_\star &\leq \int_{\tau=0}^1 (M_0 + M_1 \sup_{\theta \in [0,1]} \|\nabla f(\theta\hat{x} + (1-\theta)y)\|_\star) \|\hat{x} - y\| \, d\tau \\
&\leq \frac{M_0 \|x - y\|^2}{2} \\
&\quad + M_1 \|x - y\|^2 \int_{\tau=0}^1 \tau \sup_{\theta \in [0,1]} \|\nabla f(\theta\hat{x} + (1-\theta)y) - \nabla f(y) + \nabla f(y)\|_\star) d\tau \\
&\leq \frac{M_0 \|x - y\|^2}{2} + M_1 \|x - y\|^2 \int_{\tau=0}^1 \tau \sup_{\theta \in [0,1]} \|\nabla f(\theta\hat{x} + (1-\theta)y) - \nabla f(y)\|_\star) d\tau \\
&\quad + M_1 \|x - y\|^2 \int_{\tau=0}^1 \tau \sup_{\theta \in [0,1]} \|\nabla f(y)\|_\star \, d\tau \\
&= \frac{M_0 + M_1 \|\nabla f(y)\|}{2} \|x - y\|^2 \\
&\quad + M_1 \|x - y\|^2 \int_{\tau=0}^1 \tau \sup_{\theta \in [0,1]} \|\nabla f(\theta\hat{x} + (1-\theta)y) - \nabla f(y)\|_\star) d\tau,
\end{aligned}
$$

where $\hat{x} = x + \tau(y - x)$.

Next, from Assumption 3,

$$
\begin{aligned}
\|\nabla f(\theta\hat{x} + (1-\theta)y) - \nabla f(y)\|_\star &\leq (L_0 + L_1 \|\nabla f(y)\|_\star)\exp(\tau\theta L_1 \|x - y\|)\tau\theta \|x - y\| \\
&\overset{\theta \leq 1}{\leq} (L_0 + L_1 \|\nabla f(y)\|_\star)\exp(\tau L_1 \|x - y\|)\tau \|x - y\|.
\end{aligned}
$$

Therefore, by plugging this result into the upper-bound for $\left\|\nabla f(x) - \nabla f(y) - \nabla^2 f(y)(x - y)\right\|_\star$, we complete the proof.

$\square$

## C    PROOF OF THEOREM 1

We prove Theorem 1 in the following steps.

**Step 1) Proving the descent inequality.**    We follow the proof arguments from Lemma D.1. of Pethick et al. (2025a). By Assumption 3, and by the fact that $x_{k+1} = x_k + \text{lmo}(m_k)$,

$$
\begin{aligned}
f(x_{k+1}) \quad \leq \quad & f(x_k) + \langle \nabla f(x_k), \text{lmo}(m_k) \rangle \\
& + \frac{L_0 + L_1 \left\| \nabla f(x_k) \right\|_\star}{2} \exp(L_1 \left\| x_{k+1} - x_k \right\|) \left\| \text{lmo}(m_k) \right\|^2 \\
= \quad & f(x_k) + \langle \nabla f(x_k) - m_k, \text{lmo}(m_k) \rangle + \langle m_k, \text{lmo}(m_k) \rangle \\
& + \frac{L_0 + L_1 \left\| \nabla f(x_k) \right\|_\star}{2} \exp(L_1 \left\| x_{k+1} - x_k \right\|) \left\| \text{lmo}(m_k) \right\|^2 .
\end{aligned}
$$

By Cauchy-Schwartz inequality,

$$
\begin{aligned}
f(x_{k+1}) \quad \leq \quad & f(x_k) + \left\| \nabla f(x_k) - m_k \right\|_\star \left\| \text{lmo}(m_k) \right\| + \langle m_k, \text{lmo}(m_k) \rangle \\
& + \frac{L_0 + L_1 \left\| \nabla f(x_k) \right\|_\star}{2} \exp(L_1 \left\| x_{k+1} - x_k \right\|) \left\| \text{lmo}(m_k) \right\|^2 .
\end{aligned}
$$

By the fact that $\left\| x_{k+1} - x_k \right\| = \left\| \text{lmo}(m_k) \right\| \leq \eta_k$,

$$
f(x_{k+1}) \leq f(x_k) + \eta_k \left\| \nabla f(x_k) - m_k \right\|_\star + \langle m_k, \text{lmo}(m_k) \rangle + \frac{L_0 + L_1 \left\| \nabla f(x_k) \right\|_\star}{2} \exp(L_1 \eta_k) \eta_k^2.
$$

Since for all $u \in \mathcal{X}$

$$
\left\| u \right\|_\star = \max_{v : \left\| v \right\| \leq 1} \langle u, v \rangle = \max_{v : \left\| v \right\| \leq \eta} \left\langle u, \frac{1}{\eta} v \right\rangle = - \left\langle u, \frac{1}{\eta} \text{lmo}(u) \right\rangle,
$$

we obtain

$$
\langle m_k, \text{lmo}(m_k) \rangle = \eta_k \left\langle m_k, \frac{1}{\eta_k} \text{lmo}(m_k) \right\rangle = -\eta_k \left\| m_k \right\|_\star.
$$

Therefore,

$$
f(x_{k+1}) \leq f(x_k) + \eta_k \left\| \nabla f(x_k) - m_k \right\|_\star - \eta_k \left\| m_k \right\|_\star + \frac{L_0 + L_1 \left\| \nabla f(x_k) \right\|_\star}{2} \exp(L_1 \eta_k) \eta_k^2.
$$

By the triangle inequality, i.e. $\left\| a \right\|_\star \geq \left\| b \right\|_\star - \left\| a - b \right\|_\star$ for $a, b \in \mathbb{R}^d$,

$$
f(x_{k+1}) \leq f(x_k) + 2\eta_k \left\| \nabla f(x_k) - m_k \right\|_\star - \eta_k \left\| \nabla f(x_k) \right\|_\star + \frac{L_0 + L_1 \left\| \nabla f(x_k) \right\|_\star}{2} \exp(L_1 \eta_k) \eta_k^2.
$$

Finally, by summing the inequality over $k = 0, 1, \ldots, K$ and by re-arranging the terms,

$$
\begin{aligned}
\sum_{k=0}^{K} \eta_k \varphi_k \left\| \nabla f(x_k) \right\|_\star \quad \overset{(a)}{=} \quad & \sum_{k=0}^{K} \eta_k \left( 1 - \exp(L_1 \eta_k) \frac{L_1 \eta_k}{2} \right) \left\| \nabla f(x_k) \right\|_\star \\
\leq \quad & \sum_{k=0}^{K} f(x_k) - f(x_{k+1}) + 2 \sum_{k=0}^{K} \eta_k \left\| \nabla f(x_k) - m_k \right\|_\star + \frac{L_0}{2} \sum_{k=0}^{K} \exp(L_1 \eta_k) \eta_k^2 \\
\overset{(b)}{\leq} \quad & (f(x_0) - f_{\inf}) + 2 \sum_{k=0}^{K} \eta_k \left\| \nabla f(x_k) - m_k \right\|_\star + \frac{L_0}{2} \sum_{k=0}^{K} \exp(L_1 \eta_k) \eta_k^2 \\
\overset{(c)}{=} \quad & \Delta + 2 \sum_{k=0}^{K} \eta_k \left\| \nabla f(x_k) - m_k \right\|_\star + \frac{L_0}{2} \sum_{k=0}^{K} \exp(L_1 \eta_k) \eta_k^2, \quad (8)
\end{aligned}
$$

where we reach $(a)$ by defining $\varphi_k := (1 - \exp(L_1 \eta_k)^{L_1 \eta_k}/2)$, $(b)$ by using the fact that $f(x) \geq f_{\inf}$, and $(c)$ by denoting $\Delta := f(x_0) - f_{\inf}$.

**Step 2) Bounding the error term.** Next, we bound $\|e_{k+1}\|_\star$, where $e_{k+1} := \nabla f(x_{k+1}) - m_{k+1}$. From the definition of $e_k$,

$$
\begin{aligned}
e_{k+1} &= m_{k+1} - \nabla f(x_{k+1}) \\
&\overset{(a)}{=} (1-\alpha_k)[m_k + \nabla^2 f_{\xi_k}(\hat{x}_{k+1})(x_{k+1} - x_k) - \nabla f(x_{k+1})] \\
&\qquad + \alpha_k[\nabla f_{\xi_k}(x_{k+1}) - \nabla f(x_{k+1})] \\
&\overset{(b)}{=} (1-\alpha_k)e_k + (1-\alpha_k)W_{k+1} + \alpha_k V_{k+1},
\end{aligned}
\tag{9}
$$

where we reach $(a)$ by equation 7, and $(b)$ by denoting $W_{k+1} := \nabla^2 f_{\xi_{k+1}}(\hat{x}_{k+1})(x_{k+1} - x_k) - (\nabla f(x_{k+1}) - \nabla f(x_k))$ and $V_{k+1} := \nabla f_{\xi_{k+1}}(x_{k+1}) - \nabla f(x_{k+1})$.

By recursively applying this inequality,

$$
e_{k+1} = \prod_{t=0}^{k}(1-\alpha_t)e_0 + \sum_{t=0}^{k}\left(\prod_{j=t+1}^{k}(1-\alpha_j)\right)(1-\alpha_t)W_{t+1} + \sum_{t=0}^{k}\left(\prod_{j=t+1}^{k}(1-\alpha_j)\right)\alpha_t V_{t+1}.
$$

Next, by taking $\|\cdot\|_\star$, and by taking the expectation,

$$
\begin{aligned}
\mathrm{E}\left[\|e_{k+1}\|_\star\right] &\overset{(a)}{\leq} \|e_0\|_\star \prod_{t=0}^{k}(1-\alpha_t) + \mathrm{E}\left[\left\|\sum_{t=0}^{k}\left(\prod_{j=t+1}^{k}(1-\alpha_j)\right)(1-\alpha_t)W_{t+1}\right\|_\star\right] \\
&\quad + \mathrm{E}\left[\left\|\sum_{t=0}^{k}\left(\prod_{j=t+1}^{k}(1-\alpha_j)\right)\alpha_t V_{t+1}\right\|_\star\right] \\
&\overset{(b)}{\leq} \|e_0\|_\star \prod_{t=0}^{k}(1-\alpha_t) + \sqrt{\mathrm{E}\left[\left\|\sum_{t=0}^{k}\left(\prod_{j=t+1}^{k}(1-\alpha_j)\right)(1-\alpha_t)W_{t+1}\right\|_\star^2\right]} \\
&\quad + \sqrt{\mathrm{E}\left[\left\|\sum_{t=0}^{k}\left(\prod_{j=t+1}^{k}(1-\alpha_j)\right)\alpha_t V_{t+1}\right\|_\star^2\right]} \\
&\overset{(c)}{\leq} \|e_0\|_\star \prod_{t=0}^{k}(1-\alpha_t) + \bar{\rho}\underbrace{\sqrt{\mathrm{E}\left[\left\|\sum_{t=0}^{k}\left(\prod_{j=t+1}^{k}(1-\alpha_j)\right)(1-\alpha_t)W_{t+1}\right\|_2^2\right]}}_{:=①} \\
&\quad + \bar{\rho}\underbrace{\sqrt{\mathrm{E}\left[\left\|\sum_{t=0}^{k}\left(\prod_{j=t+1}^{k}(1-\alpha_j)\right)\alpha_t V_{t+1}\right\|_2^2\right]}}_{:=②},
\end{aligned}
$$

where $(a)$ follows from the triangle inequality, $(b)$ results from Jensen's inequality, and $(c)$ obtains from equation 2.

To bound $\mathrm{E}\left[\|e_{k+1}\|_\star\right]$, we must bound ① and ②. First, we bound ②. By Assumption 1, i.e. $\mathrm{E}_\xi\left[\nabla f_\xi(x)\right] = \nabla f(x)$ and $\mathrm{E}_\xi\left[\|\nabla f_\xi(x) - \nabla f(x)\|_2^2\right] \leq \sigma_g^2$, we can prove that $\mathrm{E}_{\xi_k}\left[V_k\right] = 0$, and that $\mathrm{E}_{\xi_j}\left[\langle V_j, V_i\rangle\right] = \langle \mathrm{E}_{\xi_j}\left[V_j\right], V_i\rangle = 0$ for $i, j \in \mathbb{N}$ and $i < j$. Therefore,

$$
② = \sum_{t=0}^{k}\left(\prod_{j=t+1}^{k}(1-\alpha_j)^2\right)\alpha_t^2 \mathrm{E}\left[\|V_t\|_2^2\right] \leq \sigma_g^2 \sum_{t=0}^{k}\left(\prod_{j=t+1}^{k}(1-\alpha_j)^2\right)\alpha_t^2.
$$

Plugging this result into the main inequality yields

$$
\mathrm{E}\left[\|e_{k+1}\|_\star\right] \leq \|e_0\|_\star \prod_{t=0}^{k}(1-\alpha_t) + \bar{\rho}\,①  + \bar{\rho}\sigma_g \sqrt{\sum_{t=0}^{k}\left(\prod_{j=t+1}^{k}(1-\alpha_j)^2\right)\alpha_t^2}.
$$

Second, we bound ①. By Assumption 1, i.e. $\mathrm{E}_\xi\left[\nabla^2 f_\xi(x)\right] = \nabla^2 f(x)$, we can prove that

$$
\begin{aligned}
\mathrm{E}_{\xi_k, b_k}\left[W_k\right] &= \mathrm{E}_{\xi_k, b_k}\left[\nabla^2 f_{\xi_k}(b_k x_k + (1-b_k)x_{k-1})(x_k - x_{k-1}) - (\nabla f(x_k) - \nabla f(x_{k-1}))\right] \\
&= \mathrm{E}_{\xi_k}\left[\int_{b=0}^{1}\nabla^2 f_{\xi_k}(bx_k + (1-b)x_{k-1})(x_k - x_{k-1}) - (\nabla f(x_k) - \nabla f(x_{k-1}))\right] \\
&= \int_{b=0}^{1}\nabla^2 f(bx_k + (1-b)x_{k-1})(x_k - x_{k-1})db - (\nabla f(x_k) - \nabla f(x_{k-1})) \\
&= 0,
\end{aligned}
$$

and that $\mathrm{E}_{\xi_j, b_j}\left[\langle W_j, W_i\rangle\right] = \langle \mathrm{E}_{\xi_j, b_j}\left[W_j\right], W_i\rangle = 0$ for $i, j \in \mathbb{N}$ and $i < j$. Thus,

$$
\begin{aligned}
\mathrm{E}\left[\|e_{k+1}\|_\star\right] &\leq \|e_0\|_\star \prod_{t=0}^{k}(1-\alpha_t) + \bar{\rho}\sqrt{\sum_{t=0}^{k}\left(\prod_{j=t+1}^{k}(1-\alpha_j)^2\right)(1-\alpha_t)^2 \mathrm{E}\left[\|W_{t+1}\|_2^2\right]} \\
&\quad + \bar{\rho}\sigma_g\sqrt{\sum_{t=0}^{k}\left(\prod_{j=t+1}^{k}(1-\alpha_j)^2\right)\alpha_t^2}.
\end{aligned}
$$

**Step 3) Bounding** $\mathrm{E}\left[\|W_{k+1}\|_2^2\right]$. Next, we bound $\mathrm{E}\left[\|W_{k+1}\|_2^2\right]$. By the fact that $W_{k+1} = \nabla^2 f_{\xi_{k+1}}(\hat{x}_{k+1})(x_{k+1} - x_k) - (\nabla f(x_{k+1}) - \nabla f(x_k))$, and that $\|x + y + z\|_2^2 \leq 3\|x\|_2^2 + 3\|y\|_2^2 + 3\|z\|_2^2$ for $x, y, z \in \mathbb{R}^d$,

$$
\begin{aligned}
\mathrm{E}\left[\|W_{k+1}\|_2^2\right] &\leq 3\mathrm{E}\left[\|\nabla f(x_{k+1}) - \nabla f(x_k)\|_2^2\right] + 3\mathrm{E}\left[\left\|\nabla^2 f(\hat{x}_{k+1})(x_{k+1} - x_k)\right\|_2^2\right] \\
&\quad + 3\mathrm{E}\left[\left\|(\nabla^2 f_{\xi_{k+1}}(\hat{x}_{k+1}) - \nabla^2 f(\hat{x}_{k+1}))(x_{k+1} - x_k)\right\|_2^2\right].
\end{aligned}
$$

Next, by Assumption 1,

$$
\begin{aligned}
\mathrm{E}\left[\|W_{k+1}\|_2^2\right] &\leq 3\mathrm{E}\left[\|\nabla f(x_{k+1}) - \nabla f(x_k)\|_2^2\right] + 3\mathrm{E}\left[\left\|\nabla^2 f(\hat{x}_{k+1})(x_{k+1} - x_k)\right\|_2^2\right] \\
&\quad + 3\sigma_H^2 \mathrm{E}\left[\|x_{k+1} - x_k\|_2^2\right] \\
&\overset{(a)}{\leq} \frac{3}{\underline{\rho}^2}\mathrm{E}\left[\|\nabla f(x_{k+1}) - \nabla f(x_k)\|_\star^2\right] + \frac{3}{\underline{\rho}^2}\mathrm{E}\left[\left\|\nabla^2 f(\hat{x}_{k+1})(x_{k+1} - x_k)\right\|_\star^2\right] \\
&\quad + \frac{3\sigma_H^2}{\underline{\theta}^2}\mathrm{E}\left[\|x_{k+1} - x_k\|^2\right] \\
&\leq \frac{3}{\underline{\rho}^2}\mathrm{E}\left[\|\nabla f(x_{k+1}) - \nabla f(x_k)\|_\star^2\right] + \frac{3}{\underline{\rho}^2}\mathrm{E}\left[\left\|\nabla^2 f(\hat{x}_{k+1})\right\|_{\mathrm{op}}^2 \|x_{k+1} - x_k\|^2\right] \\
&\quad + \frac{3\sigma_H^2}{\underline{\theta}^2}\mathrm{E}\left[\|x_{k+1} - x_k\|^2\right]
\end{aligned}
$$

where in $(a)$ we used equation 2.

Next, by the twice-differentiability of $f$, from Assumption 3, and from Proposition 3.2. and Theorem 1 of Chen et al. (2023), we can prove, respectively, that

$$
\|\nabla f(x_{k+1}) - \nabla f(x_k)\|_\star^2 \leq (L_0 + L_1\|\nabla f(x_k)\|_\star)^2 \exp(2L_1\|x_{k+1} - x_k\|)\|x_{k+1} - x_k\|^2, \quad \text{and}
$$

$$
\left\|\nabla^2 f(x)\right\|_{\mathrm{op}} := \sup_{u \neq 0}\frac{\left\|\nabla^2 f(x)u\right\|_\star}{\|u\|} = L_0 + L_1\|\nabla f(x)\|_\star,
$$

we obtain

$$
\begin{aligned}
\mathrm{E}\left[\|W_{k+1}\|_2^2\right] \;\leq\; & \frac{3}{\rho^2}\mathrm{E}\left[(L_0 + L_1\|\nabla f(x_k)\|_\star)^2 \exp(2L_1\|x_{k+1}-x_k\|)\|x_{k+1}-x_k\|^2\right] \\
& + \frac{3}{\rho^2}\mathrm{E}\left[(L_0 + L_1\|\nabla f(\hat{x}_{k+1})\|_\star)^2\|x_{k+1}-x_k\|^2\right] + \frac{3\sigma_H^2}{\theta^2}\mathrm{E}\left[\|x_{k+1}-x_k\|^2\right].
\end{aligned}
$$

By the fact that $\|x_{k+1}-x_k\| \leq \eta_k$,

$$
\begin{aligned}
\mathrm{E}\left[\|W_{k+1}\|_2^2\right] \;\leq\; & \frac{3}{\rho^2}(L_0 + L_1\|\nabla f(x_k)\|_\star)^2 \exp(2L_1\eta_k)\eta_k^2 \\
& + \frac{3}{\rho^2}\eta_k^2\mathrm{E}\left[(L_0 + L_1\|\nabla f(\hat{x}_{k+1})\|_\star)^2\right] + \frac{3\sigma_H^2}{\theta^2}\eta_k^2.
\end{aligned}
$$

Next, since

$$
\begin{aligned}
(L_0 + L_1\|\nabla f(\hat{x}_{k+1})\|_\star)^2 \;\leq\; & (L_0 + L_1\|\nabla f(x_k)\|_\star + L_1\|\nabla f(\hat{x}_{k+1}) - \nabla f(x_k)\|_\star)^2 \\
\;\leq\; & 2(L_0 + L_1\|\nabla f(x_k)\|_\star)^2 + 2L_1^2\|\nabla f(\hat{x}_{k+1}) - \nabla f(x_k)\|_\star^2 \\
\;\leq\; & 2(L_0 + L_1\|\nabla f(x_k)\|_\star)^2 \\
& + 2L_1^2(L_0 + L_1\|\nabla f(x_k)\|_\star)^2 \exp(2L_1\|\hat{x}_{k+1}-x_k\|)\|\hat{x}_{k+1}-x_k\|^2,
\end{aligned}
$$

we obtain

$$
\begin{aligned}
\mathrm{E}\left[\|W_{k+1}\|_2^2\right] \;\leq\; & \frac{3}{\rho^2}(L_0 + L_1\|\nabla f(x_k)\|_\star)^2(\exp(2L_1\eta_k) + 2)\eta_k^2 + \frac{3\sigma_H^2}{\theta^2}\eta_k^2 \\
& + \frac{6}{\rho^2}L_1^2\eta_k^2(L_0 + L_1\|\nabla f(x_k)\|_\star)^2\mathrm{E}\left[\exp(2L_1\|\hat{x}_{k+1}-x_k\|)\|\hat{x}_{k+1}-x_k\|^2\right] \\
\;\overset{(a)}{\leq}\; & \frac{3}{\rho^2}(L_0 + L_1\|\nabla f(x_k)\|_\star)^2(\exp(2L_1\eta_k) + 2)\eta_k^2 + \frac{3\sigma_H^2}{\theta^2}\eta_k^2 \\
& + \frac{6}{\rho^2}L_1^2\eta_k^2(L_0 + L_1\|\nabla f(x_k)\|)^2\mathrm{E}\left[\exp(2L_1 b_k\|x_{k+1}-x_k\|)b_k^2\|x_{k+1}-x_k\|^2\right],
\end{aligned}
$$

where $(a)$ results from the definition of $\hat{x}_{k+1}$. Next, by the fact that $\|x_{k+1}-x_k\| \leq \eta_k$,

$$
\begin{aligned}
\mathrm{E}\left[\|W_{k+1}\|_2^2\right] \;\leq\; & \frac{3}{\rho^2}(L_0 + L_1\|\nabla f(x_k)\|_\star)^2(\exp(2L_1\eta_k) + 2)\eta_k^2 \\
& + \frac{6}{\rho^2}L_1^2\eta_k^2(L_0 + L_1\|\nabla f(x_k)\|)^2\mathrm{E}\left[\exp(2L_1 b_{k+1}\eta_k)b_{k+1}^2\eta_k^2\right] + \frac{3\sigma_H^2}{\theta^2}\eta_k^2 \\
\;=\; & \frac{3}{\rho^2}(L_0 + L_1\|\nabla f(x_k)\|_\star)^2(\exp(2L_1\eta_k) + 2)\eta_k^2 \\
& + \frac{6}{\rho^2}L_1^2\eta_k^2(L_0 + L_1\|\nabla f(x_k)\|)^2\eta_k^2\int_0^1 \exp(2L_1 b\eta_k)b^2 db \\
& + \frac{3\sigma_H^2}{\theta^2}\eta_k^2.
\end{aligned}
$$

Since

$$
\int \exp(a_k z)z^2 dz = \frac{1}{a_k}z^2\exp(a_k z) - \frac{2}{a_k}\int z\exp(a_k z)dz,
$$

we obtain $\int_{z=0}^1 \exp(a_k z)z^2 dz \leq \frac{1}{a_k}\exp(a_k)$. Therefore,

$$
\begin{aligned}
\mathrm{E}\left[\|W_{k+1}\|_2^2\right] \;\leq\; & \frac{3}{\rho^2}(L_0 + L_1\|\nabla f(x_k)\|_\star)^2(\exp(2L_1\eta_k) + 2)\eta_k^2 \\
& + \frac{6}{\rho^2}L_1^2\eta_k^2(L_0 + L_1\|\nabla f(x_k)\|)^2\eta_k^2\frac{1}{2L_1\eta_k}\exp(2L_1\eta_k) + \frac{3\sigma_H^2}{\theta^2}\eta_k^2.
\end{aligned}
$$

By re-arranging the terms, and by the fact that $(a + b)^2 \leq 2a^2 + 2b^2$ for $a, b \in \mathbb{R}$,

$$
\begin{aligned}
\mathrm{E}\left[\|W_{k+1}\|_2^2\right] &\leq \frac{3}{\rho^2}(L_0 + L_1 \|\nabla f(x_k)\|_\star)^2 \left(\exp(2L_1\eta_k) + \eta_k L_1 \exp(2L_1\eta_k) + 2\right)\eta_k^2 \\
&\quad + \frac{3\sigma_H^2}{\theta^2}\eta_k^2 \\
&\leq \frac{6}{\rho^2}L_1^2 \|\nabla f(x_k)\|_\star^2 \left(\exp(2L_1\eta_k) + \eta_k L_1 \exp(2L_1\eta_k) + 2\right)\eta_k^2 \\
&\quad + \frac{6}{\rho^2}L_0^2 \left((\exp(2L_1\eta_k) + \eta_k L_1 \exp(2L_1\eta_k) + 2)\eta_k^2 + \frac{3\sigma_H^2}{\theta^2}\eta_k^2.
\end{aligned}
$$

**Step 4) Plugging $\mathrm{E}\left[\|W_k\|_2^2\right]$ back into the upper-bound for $\mathrm{E}\left[\|e_{k+1}\|_\star\right]$.** By plugging $\mathrm{E}\left[\|W_k\|_2^2\right]$ back into the upper-bound for $\mathrm{E}\left[\|e_{k+1}\|_\star\right]$,

$$
\begin{aligned}
\mathrm{E}\left[\|e_{k+1}\|_\star\right] &\leq \|e_0\|_\star \prod_{t=0}^{k}(1 - \alpha_t) + \bar{\rho}\sqrt{\sum_{t=0}^{k}\left(\prod_{j=t+1}^{k}(1-\alpha_j)^2\right)(1-\alpha_t)^2(A_{t+1} + B_{t+1})} \\
&\quad + \bar{\rho}\sigma_g\sqrt{\sum_{t=0}^{k}\left(\prod_{j=t+1}^{k}(1-\alpha_j)^2\right)\alpha_t^2},
\end{aligned}
$$

where $A_{k+1} = \frac{6}{\rho^2}L_1^2(\exp(2L_1\eta_k) + \eta_k L_1 \exp(2L_1\eta_k) + 2)\eta_k^2 \mathrm{E}\left[\|\nabla f(x_k)\|_\star^2\right]$ and $B_{k+1} = \frac{6}{\rho^2}L_0^2(\exp(2L_1\eta_k) + \eta_k L_1 \exp(2L_1\eta_k) + 2)\eta_k^2 + \frac{3\sigma_H^2}{\theta^2}\eta_k^2$.

**Step 5) Deriving the convergence bound under constant tuning parameters.** If $\eta_k = \eta$ and $\alpha_k = \alpha$, then

$$
\begin{aligned}
\mathrm{E}\left[\|e_{k+1}\|_\star\right] &\leq (1-\alpha)^{k+1}\|e_0\|_\star + \bar{\rho}\sqrt{\sum_{t=0}^{k}(1-\alpha)^{2(k-t+1)}(A_{t+1} + B)} \\
&\quad + \bar{\rho}\sigma_g\sqrt{\sum_{t=0}^{k}(1-\alpha)^{2(k-t)}\alpha^2} \\
&\leq (1-\alpha)^{k+1}\|e_0\|_\star + \bar{\rho}\sqrt{\sum_{t=0}^{k}(1-\alpha)^{2(k-t+1)}A_{t+1}} + \bar{\rho}\sqrt{\sum_{t=0}^{k}(1-\alpha)^{2(k-t+1)}B} \\
&\quad + \bar{\rho}\sigma_g\sqrt{\sum_{t=0}^{k}(1-\alpha)^{2(k-t)}\alpha^2},
\end{aligned}
$$

where $A_{k+1} = c\eta^2 \|\nabla f(x_k)\|_\star^2$, $B = \frac{6}{\rho^2}L_0^2(\exp(2L_1\eta) + L_1\eta \exp(2L_1\eta) + 2)\eta^2 + \frac{3\sigma_H^2}{\theta^2}\eta^2$, and $c = \frac{6}{\rho^2}L_1^2(\exp(2L_1\eta) + L_1\eta \exp(2L_1\eta) + 2)$.

Next, since

$$
\sum_{t=0}^{k-1}(1-\alpha)^{2(k-t+1)} \leq \sum_{j=0}^{\infty}((1-\alpha)^2)^j = \frac{1}{1-(1-\alpha)^2} = \frac{1}{\alpha(2-\alpha)} \overset{\alpha\in[0,1]}{\leq} \frac{1}{\alpha},
$$

and

$$
\sum_{t=0}^{k-1}(1-\alpha)^{2(k-t)}\alpha^2 \leq \alpha^2\sum_{j=0}^{\infty}((1-\alpha)^2)^j = \frac{\alpha^2}{1-(1-\alpha)^2} = \frac{\alpha}{2-\alpha} \overset{\alpha\in[0,1]}{\leq} \alpha,
$$

we obtain

$$
\begin{aligned}
\mathrm{E}\left[\|e_{k+1}\|_\star\right] &\overset{(a)}{\leq} (1-\alpha)^{k+1}\|e_0\|_\star + \bar{\rho}\sqrt{\sum_{t=0}^{k}(1-\alpha)^{2(k-t)}(1-\alpha)^2 A_{t+1}} + \bar{\rho}\frac{\eta}{\sqrt{\alpha}}\hat{B} \\
&\quad + \bar{\rho}\sqrt{\alpha}\sigma_g \\
&\overset{(b)}{\leq} (1-\alpha)^{k+1}\|e_0\|_\star + \bar{\rho}\sqrt{\sum_{t=0}^{k}(1-\alpha)^{2(k-t+1)}c\eta^2\|\nabla f(x_t)\|_\star^2} + \bar{\rho}\frac{\eta}{\sqrt{\alpha}}\hat{B} \\
&\quad + \bar{\rho}\sqrt{\alpha}\sigma_g \\
&\leq (1-\alpha)^k\|e_0\|_\star + \bar{\rho}\hat{c}\eta\sum_{t=0}^{k}(1-\alpha)^{(k-t+1)}\|\nabla f(x_t)\|_\star + \bar{\rho}\frac{\eta}{\sqrt{\alpha}}\hat{B} \\
&\quad + \bar{\rho}\sqrt{\alpha}\sigma_g,
\end{aligned}
$$

where we reach $(a)$ by denoting $\hat{c} = \frac{3}{\rho}L_1(\exp(L_1\eta) + \sqrt{L_1\eta}\exp(L_1\eta) + 2)$ and $\hat{B} = \frac{3}{\rho}L_0(\exp(L_1\eta) + \sqrt{L_1\eta}\exp(L_1\eta) + 2) + \frac{2\sigma_H}{\theta}$, and $(b)$ by using the condition that $\alpha \in [0,1]$ and the definition of $A_{t+1}$.

Plugging the above result into the main descent inequality with $\eta_k = \eta$ and $\alpha_k = \alpha$ and denoting $\varphi = \left(1 - \exp(L_1\eta)\frac{L_1\eta}{2}\right)$, we obtain

$$
\begin{aligned}
\sum_{k=0}^{K}\eta\varphi\mathrm{E}\left[\|\nabla f(x_k)\|_\star\right] &\leq \Delta + 2\sum_{k=0}^{K}\eta\mathrm{E}\left[\|e_k\|_\star\right] + \frac{L_0}{2}\sum_{k=0}^{K}\exp(L_1\eta)\eta^2 \\
&\leq \Delta + 2\eta\|e_0\|_\star\sum_{k=0}^{K}(1-\alpha)^k + \frac{L_0}{2}\exp(L_1\eta)\eta^2(K+1) \\
&\quad + 2\bar{\rho}\hat{c}\eta^2\sum_{k=0}^{K}\sum_{t=0}^{k-1}(1-\alpha)^{(k-t)}\mathrm{E}\left[\|\nabla f(x_t)\|_\star\right] \\
&\quad + 2\bar{\rho}\frac{\eta^2}{\sqrt{\alpha}}\hat{B}(K+1) + 2\bar{\rho}\eta\sqrt{\alpha}\sigma_g(K+1).
\end{aligned}
$$

By the fact that $\sum_{k=0}^{K}(1-\alpha)^k \leq \sum_{k=0}^{\infty}(1-\alpha)^k = \frac{1}{\alpha}$,

$$
\begin{aligned}
\sum_{k=0}^{K}\eta\varphi\mathrm{E}\left[\|\nabla f(x_k)\|_\star\right] &\leq \Delta + \frac{2\eta\|e_0\|_\star}{\alpha} + 2\bar{\rho}\hat{c}\eta^2\sum_{k=0}^{K}\sum_{t=0}^{k-1}(1-\alpha)^{(k-t)}\mathrm{E}\left[\|\nabla f(x_t)\|_\star\right] \\
&\quad + 2\bar{\rho}\frac{\eta^2}{\sqrt{\alpha}}\hat{B}(K+1) + 2\bar{\rho}\eta\sqrt{\alpha}\sigma_g(K+1) + \frac{L_0}{2}\exp(L_1\eta)\eta^2(K+1).
\end{aligned}
$$

Next, since

$$
\begin{aligned}
\sum_{k=0}^{K}\sum_{t=0}^{k-1}(1-\alpha)^{k-t}\mathrm{E}\left[\|\nabla f(x_t)\|_\star\right] &\leq \sum_{k=0}^{K}\sum_{t=0}^{k}(1-\alpha)^{k-t}\mathrm{E}\left[\|\nabla f(x_t)\|_\star\right] \\
&= \sum_{t=0}^{K}(\sum_{k=t}^{K}(1-\alpha)^{k-t})\mathrm{E}\left[\|\nabla f(x_t)\|_\star\right] \\
&\leq \sum_{t=0}^{K}(\sum_{k=0}^{\infty}(1-\alpha)^k)\mathrm{E}\left[\|\nabla f(x_t)\|_\star\right] \\
&= \frac{1}{\alpha}\sum_{k=0}^{K}\mathrm{E}\left[\|\nabla f(x_k)\|_\star\right],
\end{aligned}
$$

we obtain

$$\sum_{k=0}^{K} \eta(\varphi - 2\bar{\rho}\hat{c}\frac{\eta}{\alpha})\mathrm{E}\left[\|\nabla f(x_k)\|_\star\right] \leq \Delta + \frac{2\eta\|e_0\|_\star}{\alpha} + 2\bar{\rho}\frac{\eta^2}{\sqrt{\alpha}}\hat{B}(K+1)$$

$$+2\bar{\rho}\eta\sqrt{\alpha}\sigma_g(K+1) + \frac{L_0}{2}\exp(L_1\eta)\eta^2(K+1).$$

**Step 6) Choosing tuning parameters.** If $\eta \leq \frac{\alpha}{80L_1}\left(\frac{\bar{\rho}}{\rho}\right)^{-1}$, then

$$\frac{\eta}{2}\sum_{k=0}^{K}\mathrm{E}\left[\|\nabla f(x_k)\|_\star\right] \leq \Delta + \frac{2\eta\|e_0\|_\star}{\alpha} + 2\bar{\rho}\frac{\eta^2}{\sqrt{\alpha}}\hat{B}(K+1)$$

$$+2\bar{\rho}\eta\sqrt{\alpha}\sigma_g(K+1) + \frac{L_0}{2}\exp(L_1\eta)\eta^2(K+1).$$

Therefore,

$$\min_{k\in\{0,1,\ldots,K\}}\mathrm{E}\left[\|\nabla f(x_k)\|_\star\right] \leq \frac{1}{K+1}\sum_{k=0}^{K}\mathrm{E}\left[\|\nabla f(x_k)\|_\star\right]$$

$$\leq \frac{2\Delta}{\eta(K+1)} + \frac{4\|e_0\|_\star}{\alpha(K+1)} + 4\bar{\rho}\frac{\eta}{\sqrt{\alpha}}\hat{B}$$

$$+4\bar{\rho}\sqrt{\alpha}\sigma_g + L_0\exp(L_1\eta)\eta.$$

If $\eta = \frac{\hat{\eta}}{(K+1)^{2/3}}$ with $\hat{\eta} = \frac{1}{80L_1}\left(\frac{\bar{\rho}}{\rho}\right)^{-1}$, and $\alpha = \frac{1}{(K+1)^{2/3}}$, then

$$\min_{k\in\{0,1,\ldots,K\}}\mathrm{E}\left[\|\nabla f(x_k)\|_\star\right] \leq \frac{2\Delta}{\hat{\eta}(K+1)^{1/3}} + \frac{4\|e_0\|_\star}{(K+1)^{1/3}} + \frac{4\bar{\rho}\hat{\eta}\hat{B}}{(K+1)^{1/3}}$$

$$+4\bar{\rho}\frac{1}{(K+1)^{1/3}}\sigma_g + L_0\exp(L_1\hat{\eta})\frac{\hat{\eta}}{(K+1)^{2/3}}.$$

# D PROOF OF THEOREM 2

We prove the result in the following steps.

**Step 1) Proving the descent inequality.** By following the proof arguments in Step 1) of Theorem 1 (see equation 8), we have

$$\sum_{k=0}^{K} \eta_k \varphi_k \left\| \nabla f(x_k) \right\|_\star \leq \Delta + 2 \sum_{k=0}^{K} \eta_k \left\| \nabla f(x_k) - m_k \right\|_\star + \frac{L_0}{2} \sum_{k=0}^{K} \exp(L_1 \eta_k) \eta_k^2,$$

where $\varphi_k := (1 - \exp(L_1 \eta_k)^{L_1 \eta_k/2})$ and $\Delta := f(x_0) - f_{\inf}$.

**Step 2) Bounding the error term.** Next, we bound $\|e_k\|_\star$, where $e_k := m_k - \nabla f(x_k)$. From the definition of $e_k$,

$$\begin{aligned}
e_{k+1} &= m_{k+1} - \nabla f(x_{k+1}) \\
&= (1 - \alpha_k)e_k + (1 - \alpha_k)Z_f(x_k; x_{k+1}) + (1 - \alpha_k)\varepsilon_{k+1}^H + \alpha_k \varepsilon_{k+1}^g,
\end{aligned}$$

where $Z_f(x_k, x_{k+1}) = \nabla f(x_k) - \nabla f(x_{k+1}) + \nabla^2 f(x_{k+1})(x_{k+1} - x_k)$, $\varepsilon_{k+1}^H = (\nabla^2 f_{\xi_{k+1}}(x_{k+1}) - \nabla^2 f(x_{k+1}))(x_{k+1} - x_k)$ and $\varepsilon_{k+1}^g = \nabla f_{\xi_{k+1}}(x_{k+1}) - \nabla f(x_{k+1})$.

By recursively applying the above inequality,

$$\begin{aligned}
e_{k+1} &= \prod_{t=0}^{k}(1 - \alpha_t)e_0 + \sum_{t=0}^{k} \left( \prod_{j=t+1}^{k}(1 - \alpha_j) \right)(1 - \alpha_t)Z_f(x_t, x_{t+1}) \\
&\quad + \sum_{t=0}^{k} \left( \prod_{j=t+1}^{k}(1 - \alpha_j) \right)(1 - \alpha_t)\varepsilon_{t+1}^H + \sum_{t=0}^{k} \left( \prod_{j=t+1}^{k}(1 - \alpha_j) \right)\alpha_t \varepsilon_{t+1}^g.
\end{aligned}$$

Therefore,

$$\begin{aligned}
\mathrm{E}\left[\|e_{k+1}\|_\star\right] &\leq \mathrm{E}\left[\left\|\prod_{t=0}^{k}(1 - \alpha_t)e_0\right\|_\star\right] + \underbrace{\mathrm{E}\left[\left\|\sum_{t=0}^{k} \left( \prod_{j=t+1}^{k}(1 - \alpha_j) \right)(1 - \alpha_t))Z_f(x_t, x_{t+1})\right\|_\star\right]}_{:=③} \\
&\quad + \underbrace{\mathrm{E}\left[\left\|\sum_{t=0}^{k} \left( \prod_{j=t+1}^{k}(1 - \alpha_j) \right)(1 - \alpha_t))\varepsilon_{t+1}^H\right\|_\star\right]}_{:=④} + \underbrace{\mathrm{E}\left[\left\|\sum_{t=0}^{k} \left( \prod_{j=t+1}^{k}(1 - \alpha_j) \right)\alpha_t \varepsilon_{t+1}^g\right\|_\star\right]}_{:=⑤}.
\end{aligned}$$

Next, we bound ⑤ by using Assumption 1 and by following the proof arguments for bounding ① in Step 2) of the convergence proof for Theorem 1. Then, we have

$$\begin{aligned}
⑤ &\leq \bar{\rho}\mathrm{E}\left[\left\|\sum_{t=0}^{k} \left( \prod_{j=t+1}^{k}(1 - \alpha_j) \right)\alpha_t \varepsilon_{t+1}^g\right\|_2\right] \\
&\leq \bar{\rho}\sqrt{\mathrm{E}\left[\left\|\sum_{t=0}^{k} \left( \prod_{j=t+1}^{k}(1 - \alpha_j) \right)\alpha_t \varepsilon_{t+1}^g\right\|_2^2\right]} \\
&\leq \bar{\rho}\sigma_g\sqrt{\sum_{t=0}^{k} \left( \prod_{j=t+1}^{k}(1 - \alpha_j)^2 \right)\alpha_t^2},
\end{aligned}$$

Therefore,

$$\mathrm{E}\left[\|e_{k+1}\|_\star\right] \;\leq\; \prod_{t=0}^{k}(1-\alpha_t)\mathrm{E}\left[\|e_0\|_\star\right] + \text{③} \;+\; \text{④} \;+\; \bar{\rho}\sigma_g\sqrt{\sum_{t=0}^{k}\left(\prod_{j=t+1}^{k}(1-\alpha_j)^2\right)\alpha_t^2}.$$

Next, we bound ④.

$$\text{④} \;\leq\; \bar{\rho}\mathrm{E}\left[\left\|\sum_{t=0}^{k}\left(\prod_{j=t+1}^{k}(1-\alpha_j)\right)(1-\alpha_t))\varepsilon_{t+1}^H\right\|_2\right]$$

$$\leq\; \bar{\rho}\sqrt{\mathrm{E}\left[\left\|\sum_{t=0}^{k}\left(\prod_{j=t+1}^{k}(1-\alpha_j)\right)(1-\alpha_t))\varepsilon_{t+1}^H\right\|_2^2\right]}.$$

From the definition of $\varepsilon_{t+1}^H$ and from Assumption 1,

$$\text{④} \;\leq\; \bar{\rho}\sqrt{\sum_{t=0}^{k}\left(\prod_{j=t+1}^{k}(1-\alpha_j)^2\right)(1-\alpha_t)^2)\mathrm{E}\left[\left\|\varepsilon_{t+1}^H\right\|_2^2\right]}$$

$$\leq\; \bar{\rho}\sqrt{\sum_{t=0}^{k}\left(\prod_{j=t+1}^{k}(1-\alpha_j)^2\right)(1-\alpha_t)^2)\sigma_H^2\left\|x_{t+1}-x_t\right\|_2^2}$$

$$\leq\; \frac{\bar{\rho}}{\underline{\theta}}\sqrt{\sum_{t=0}^{k}\left(\prod_{j=t+1}^{k}(1-\alpha_j)^2\right)(1-\alpha_t)^2)\sigma_H^2\left\|x_{t+1}-x_t\right\|^2}$$

$$\leq\; \frac{\bar{\rho}}{\underline{\theta}}\eta\sigma_H\sqrt{\sum_{t=0}^{k}\left(\prod_{j=t+1}^{k}(1-\alpha_j)^2\right)(1-\alpha_t)^2}.$$

Therefore,

$$\mathrm{E}\left[\|e_{k+1}\|_\star\right] \;\leq\; \prod_{t=0}^{k}(1-\alpha_t)\mathrm{E}\left[\|e_0\|_\star\right] + \frac{\bar{\rho}}{\underline{\theta}}\eta\sigma_H\sqrt{\sum_{t=0}^{k}\left(\prod_{j=t+1}^{k}(1-\alpha_j)^2\right)(1-\alpha_t)^2}$$

$$+\; \text{③} \;+\; \bar{\rho}\sigma_g\sqrt{\sum_{t=0}^{k}\left(\prod_{j=t+1}^{k}(1-\alpha_j)^2\right)\alpha_t^2}.$$

Next, we bound ③:

$$\text{③} \;\leq\; \sum_{t=0}^{k}\left(\prod_{j=t+1}^{k}(1-\alpha_j)\right)(1-\alpha_t))\mathrm{E}\left[\|Z_f(x_t,x_{t+1})\|_\star\right].$$

From Lemma 1,

$$\|Z_f(x_t,x_{t+1})\|_\star \;\leq\; \frac{1}{2}(M_0 + M_1\|\nabla f(x_{t+1})\|_\star)\|x_t - x_{t+1}\|^2$$

$$+\frac{1}{3}M_1(L_0 + L_1\|\nabla f(x_{t+1})\|_\star)\exp(L_1\|x_t - x_{t+1}\|)\|x_t - x_{t+1}\|^3$$

$$\leq\; \frac{1}{2}(M_0 + M_1\|\nabla f(x_{t+1})\|_\star)\eta_t^2 + \frac{1}{3}M_1(L_0 + L_1\|\nabla f(x_{t+1})\|_\star)\exp(L_1\eta_t)\eta_t^3$$

$$=\; \left(\frac{M_0}{2} + \frac{M_1}{3}L_0\exp(L_1\eta_t)\right)\eta_t^2 + \left(\frac{M_1}{2} + \frac{M_1}{3}L_1\eta_t\exp(L_1\eta_t)\right)\eta_t^2\|\nabla f(x_{t+1})\|_\star.$$

Therefore,

$$
\text{③} \quad \leq \quad \sum_{t=0}^{k} \left( \prod_{j=t+1}^{k} (1-\alpha_j) \right) (1-\alpha_t)) \left( \frac{M_0}{2} + \frac{M_1}{3} L_0 \exp(L_1\eta_t) \right) \eta_t^2
$$

$$
+ \sum_{t=0}^{k} \left( \prod_{j=t+1}^{k} (1-\alpha_j) \right) (1-\alpha_t)) \left( \frac{M_1}{2} + \frac{M_1}{3} L_1\eta_t \exp(L_1\eta_t) \right) \eta_t^2 \left\| \nabla f(x_{t+1}) \right\|_\star .
$$

Plugging ③into the upper-bound for $\mathrm{E}\left[\left\|e_{k+1}\right\|_\star\right]$ yields

$$
\mathrm{E}\left[\left\|e_{k+1}\right\|_\star\right] \quad \leq \quad \prod_{t=0}^{k}(1-\alpha_t)\mathrm{E}\left[\left\|e_0\right\|_\star\right]
$$

$$
+ \sum_{t=0}^{k} \left( \prod_{j=t+1}^{k} (1-\alpha_j) \right) (1-\alpha_t) \left( \frac{M_0}{2} + \frac{M_1}{3} L_0 \exp(L_1\eta_t) \right) \eta_t^2
$$

$$
+ \sum_{t=0}^{k} \left( \prod_{j=t+1}^{k} (1-\alpha_j) \right) (1-\alpha_t) \left( \frac{M_1}{2} + \frac{M_1}{3} L_1\eta_t \exp(L_1\eta_t) \right) \eta_t^2 \left\| \nabla f(x_{t+1}) \right\|_\star
$$

$$
+ \frac{\bar{\rho}}{\underline{\theta}}\eta\sigma_H \sqrt{ \sum_{t=0}^{k} \left( \prod_{j=t+1}^{k} (1-\alpha_j)^2 \right) (1-\alpha_t)^2 } + \bar{\rho}\sigma_g \sqrt{ \sum_{t=0}^{k} \left( \prod_{j=t+1}^{k} (1-\alpha_j)^2 \right) \alpha_t^2 } .
$$

If $\eta_k = \eta$ and $\alpha_k = \alpha$, then

$$
\mathrm{E}\left[\left\|e_{k+1}\right\|_\star\right] \quad \leq \quad (1-\alpha)^{k+1} \left\|e_0\right\|_\star + \left( \frac{M_0}{2} + \frac{M_1 L_0 \exp(L_1\eta)}{3} \right) \sum_{t=0}^{k}(1-\alpha)^{k-t+1}\eta^2
$$

$$
+ \left( \frac{M_1}{2} + \frac{M_1 L_1 \exp(L_1\eta)\eta}{3} \right) \sum_{t=0}^{k}(1-\alpha)^{k-t+1}\eta^2 \left\| \nabla f(x_{t+1}) \right\|_\star
$$

$$
+ \frac{\bar{\rho}}{\underline{\theta}}\eta\sigma_H \sqrt{ \sum_{t=0}^{k}(1-\alpha)^{2(k-t+1)} } + \bar{\rho}\sigma_g \sqrt{ \sum_{t=0}^{k}(1-\alpha)^{2(k-t)}\alpha^2 } .
$$

Next, since

$$
\sum_{t=0}^{k-1}(1-\alpha)^{2(k-t+1)} \quad \leq \quad \sum_{j=0}^{\infty}((1-\alpha)^2)^j = \frac{1}{\alpha(2-\alpha)} \overset{\alpha\in[0,1]}{\leq} \frac{1}{\alpha},
$$

and

$$
\sum_{t=0}^{k-1}(1-\alpha)^{2(k-t)}\alpha^2 \quad \leq \quad \alpha^2 \sum_{j=0}^{\infty}((1-\alpha)^2)^j = \frac{\alpha^2}{1-(1-\alpha)^2} = \frac{\alpha}{2-\alpha} \overset{\alpha\in[0,1]}{\leq} \alpha,
$$

we obtain

$$
\mathrm{E}\left[\left\|e_{k+1}\right\|_\star\right] \quad \leq \quad (1-\alpha)^{k+1} \left\|e_0\right\|_\star + \left( \frac{M_0}{2} + \frac{M_1}{3} L_0 \exp(L_1\eta) \right) \frac{\eta^2}{\alpha}
$$

$$
+ \left( \frac{M_1}{2} + \frac{M_1}{3} L_1\eta \exp(L_1\eta) \right) \sum_{t=0}^{k}(1-\alpha)^{k-t+1}\eta^2 \left\| \nabla f(x_{t+1}) \right\|_\star
$$

$$
+ \frac{\bar{\rho}}{\underline{\theta}} \frac{\eta}{\sqrt{\alpha}}\sigma_H + \bar{\rho}\sqrt{\alpha}\sigma_g .
$$

Therefore, recalling $\varphi = \eta(1 - \exp(L_1\eta)^{L_1\eta/2})$, we have

$$
\begin{aligned}
\sum_{k=0}^{K} \eta\varphi \mathrm{E}\left[\|\nabla f(x_k)\|_\star\right] &\leq \Delta + \frac{L_0}{2}\sum_{k=0}^{K}\exp(L_1\eta)\eta^2 + 2\eta\sum_{k=0}^{K}\mathrm{E}\left[\|e_k\|_\star\right] \\
&\leq \Delta + \frac{L_0}{2}\exp(L_1\eta)\eta^2(K+1) + 2\eta\sum_{k=0}^{K}(1-\alpha)^k\|e_0\|_\star \\
&\quad +\eta\left(M_0 + \frac{2M_1}{3}L_0\exp(L_1\eta)\right)\frac{\eta^2}{\alpha}(K+1) \\
&\quad +\eta\left(M_1 + \frac{2M_1}{3}L_1\eta\exp(L_1\eta)\right)\sum_{k=0}^{K}\sum_{t=0}^{k-1}(1-\alpha)^{k-t}\eta^2\|\nabla f(x_{t+1})\|_\star \\
&\quad +2\eta\left(\frac{\bar{\rho}}{\underline{\theta}}\frac{\eta}{\sqrt{\alpha}}\sigma_H + \bar{\rho}\sqrt{\alpha}\sigma_g\right)(K+1),
\end{aligned}
$$

where $\Delta = \mathrm{E}\left[f(x_0) - f_{\inf}\right]$.

Next, since

$$
\sum_{k=0}^{K}(1-\alpha)^k \leq \frac{1}{\alpha}, \quad \text{and} \quad \sum_{k=0}^{K}\sum_{t=0}^{k-1}(1-\alpha)^{k-t}\|\nabla f(x_{t+1})\|_\star \leq \frac{1}{\alpha}\sum_{k=0}^{K}\|\nabla f(x_k)\|_\star,
$$

we obtain

$$
\begin{aligned}
\sum_{k=0}^{K}\eta\vartheta\mathrm{E}\left[\|\nabla f(x_k)\|_\star\right] &\leq \Delta + (K+1)\frac{L_0}{2}\exp(L_1\eta)\eta^2 + 2\frac{\eta}{\alpha}\|e_0\|_\star \\
&\quad +\eta(K+1)\left(M_0 + \frac{2M_1}{3}L_0\exp(L_1\eta)\right)\frac{\eta^2}{\alpha} \\
&\quad +2\eta\left(\frac{\bar{\rho}}{\underline{\theta}}\frac{\eta}{\sqrt{\alpha}}\sigma_H + \bar{\rho}\sqrt{\alpha}\sigma_g\right)(K+1),
\end{aligned}
$$

where $\vartheta := \left(1 - \exp(L_1\eta)^{L_1\eta/2} - (M_1 + {}^{2M_1}/3 \cdot L_1\eta\exp(L_1\eta))\frac{\eta^2}{\alpha}\right)$.

If $\eta \leq \frac{\alpha}{3}\min\left\{\frac{1}{L_1}, \frac{1}{\sqrt{M_1}}\right\}$, then $\eta \leq \min\left\{\frac{1}{3L_1}, \frac{1}{3\sqrt{M_1}}\right\}$ and

$$
\begin{aligned}
\frac{\eta}{2}\sum_{k=0}^{K}\mathrm{E}\left[\|\nabla f(x_k)\|_\star\right] &\leq \Delta + (K+1)\frac{L_0}{2}\exp(L_1\eta)\eta^2 + 2\frac{\eta}{\alpha}\|e_0\|_\star \\
&\quad +\eta(K+1)\left(M_0 + \frac{2M_1}{3}L_0\exp(L_1\eta)\right)\frac{\eta^2}{\alpha} \\
&\quad +2\eta\left(\frac{\bar{\rho}}{\underline{\theta}}\frac{\eta}{\sqrt{\alpha}}\sigma_H + \bar{\rho}\sqrt{\alpha}\sigma_g\right)(K+1).
\end{aligned}
$$

Therefore,

$$
\begin{aligned}
\frac{1}{K+1}\sum_{k=0}^{K}\mathrm{E}\left[\|\nabla f(x_k)\|_\star\right] &\leq \frac{2\Delta}{\eta(K+1)} + L_0\exp(L_1\eta)\eta + \frac{4\|e_0\|_\star}{\alpha(K+1)} \\
&\quad +2\left(M_0 + \frac{2M_1}{3}L_0\exp(L_1\eta)\right)\frac{\eta^2}{\alpha} \\
&\quad +4\left(\frac{\bar{\rho}}{\underline{\theta}}\frac{\eta}{\sqrt{\alpha}}\sigma_H + \bar{\rho}\sqrt{\alpha}\sigma_g\right).
\end{aligned}
$$

If $\eta = \hat{\eta}\alpha$ with $\hat{\eta} = \min\left\{\frac{1}{5L_1}, \frac{1}{3\sqrt{M_1}}\right\}$ and $\alpha = \frac{1}{(K+1)^{2/3}}$, then

$$
\begin{aligned}
\frac{1}{K+1}\sum_{k=0}^{K} \mathrm{E}\left[\|\nabla f(x_k)\|_\star\right] \leq\ & \frac{2\Delta}{\hat{\eta}(K+1)^{1/3}} + \frac{\hat{\eta}L_0\exp(L_1\hat{\eta})}{(K+1)^{2/3}} + \frac{4\|e_0\|_\star}{(K+1)^{1/3}} \\
& +2\left(M_0 + \frac{2M_1}{3}L_0\exp(L_1\hat{\eta})\right)\frac{\hat{\eta}^2}{(K+1)^{2/3}} \\
& +4\frac{\bar{\rho}}{\underline{\theta}}\frac{\hat{\eta}\sigma_H}{(K+1)^{1/3}} + 4\frac{\bar{\rho}\sigma_g}{(K+1)^{1/3}}.
\end{aligned}
$$

Finally, by the fact that $\min_{k\in\{0,1,\ldots,K\}}\mathrm{E}\left[\|\nabla f(x_k)\|_\star\right] \leq \frac{1}{K+1}\sum_{k=0}^{K}\mathrm{E}\left[\|\nabla f(x_k)\|_\star\right]$, we obtain the final result.

# E  LMO-BASED METHODS WITH EXTRAPOLATED MOMENTUM

In this section, we present the convergence of LMO-based methods in equation 3 that leverage extrapolated momentum in equation 4.

**Theorem 3.** Consider the problem of minimizing $f(x) = \mathrm{E}_{\xi \sim \mathcal{D}}[f_\xi(x)]$. Let $f$ be twice differentiable, and let Assumptions 1, 2, 3, and 4 hold. Then, the iterates $\{x_k\}$ generated by LMO-based methods in equation 3 that leverage extrapolated momentum in equation 4 with

$$\alpha_k = \alpha = \frac{1}{(K+1)^{4/7}}, \quad \text{and} \quad \eta_k = \eta = \frac{\hat{\eta}}{(K+1)^{5/7}},$$

where $\hat{\eta} = \min\left\{\frac{1}{3L_1}, \frac{1}{3\sqrt{M_1}}\right\}$ satisfy

$$\min_{k \in \{0,1,\dots,K\}} \mathrm{E}\left[\|\nabla f(x_k)\|_\star\right] \leq \frac{2(f(x_0 - f_{\inf})}{\hat{\eta}(K+1)^{2/7}} + \frac{\hat{\eta} L_0 \exp(L_1 \hat{\eta})}{(K+1)^{5/7}} + \frac{4\|e_0\|_\star}{(K+1)^{2/7}} + 4\frac{\bar{\rho}\sigma_g}{(K+1)^{2/7}}$$
$$+ 2\left(M_0 + \frac{2M_1}{3} L_0 \exp(L_1 \hat{\eta})\right)\frac{\hat{\eta}^2}{(K+1)^{2/7}}.$$

Theorem 3 establishes the $\mathcal{O}(1/K^{2/7})$ convergence of LMO-based methods using extrapolated momentum under relaxed smoothness with respect to the arbitrary norm, which matches the known rate obtained by Kovalev (2025, Corollary 5) under traditional smoothness with respect to the arbitrary norm.

## E.1  PROOF OF THEOREM 3

We prove the result in the following steps.

**Step 1) Proving the descent inequality.**  By following the proof arguments in Step 1) of Theorem 1 (see equation 8), we have

$$\sum_{k=0}^{K} \eta_k \varphi_k \|\nabla f(x_k)\|_\star \leq \Delta + 2\sum_{k=0}^{K} \eta_k \|\nabla f(x_k) - m_k\|_\star + \frac{L_0}{2}\sum_{k=0}^{K} \exp(L_1\eta_k)\eta_k^2,$$

where $\varphi_k := (1 - \exp(L_1\eta_k)L_1\eta_k/2)$ and $\Delta := f(x_0) - f_{\inf}$.

**Step 2) Bounding the error term.**  Next, we bound $\|e_k\|_\star$, where $e_k := m_k - \nabla f(x_k)$. From the definition of $e_k$,

$$\begin{aligned}
e_{k+1} &= m_{k+1} - \nabla f(x_{k+1}) \\
&= (1-\alpha_k)e_k + \alpha_k\left(\nabla f(y_{k+1}) - \nabla f(x_{k+1}) - \nabla^2 f(x_{k+1})(y_{k+1} - x_{k+1})\right) \\
&\quad + (1-\alpha_k)\left(\nabla f(x_k) - \nabla f(x_{k+1}) + \frac{\alpha_k}{1-\alpha_k}\nabla^2 f(x_{k+1})(y_{k+1} - x_{k+1})\right) \\
&\quad + \alpha_k\left(\nabla f(x_{k+1};\xi_{k+1}) - \nabla f(x_{k+1})\right) \\
&= (1-\alpha_k)e_k + (1-\alpha_k)Z_f(x_k;x_{k+1}) + \alpha_k Z_f(y_{k+1}, x_{k+1}) + \alpha_k \varepsilon_{k+1}^g,
\end{aligned}$$

where $Z_f(x,y) = \nabla f(x) - \nabla f(y) + \nabla^2 f(y)(x-y)$ and $\varepsilon_{k+1}^g = \nabla f(x_{k+1};\xi_{k+1}) - \nabla f(x_{k+1})$. By recursively applying the above inequality,

$$\begin{aligned}
e_{k+1} &= \prod_{t=0}^{k}(1-\alpha_t)e_0 + \sum_{t=0}^{k}\left(\prod_{j=t+1}^{k}(1-\alpha_j)\right)(1-\alpha_t)Z_f(x_t, x_{t+1}) \\
&\quad + \sum_{t=0}^{k}\left(\prod_{j=t+1}^{k}(1-\alpha_j)\right)\alpha_t Z_f(y_{t+1}, x_{t+1}) + \sum_{t=0}^{k}\left(\prod_{j=t+1}^{k}(1-\alpha_j)\right)\alpha_t \varepsilon_{t+1}^g.
\end{aligned}$$

Therefore,

$$
\mathrm{E}\left[\|e_{k+1}\|_\star\right] \leq \mathrm{E}\left[\left\|\prod_{t=0}^{k}(1-\alpha_t)e_0\right\|_\star\right] + \underbrace{\mathrm{E}\left[\left\|\sum_{t=0}^{k}\left(\prod_{j=t+1}^{k}(1-\alpha_j)\right)(1-\alpha_t)Z_f(x_t,x_{t+1})\right\|_\star\right]}_{:=\text{⑥}}
$$

$$
+ \underbrace{\mathrm{E}\left[\left\|\sum_{t=0}^{k}\left(\prod_{j=t+1}^{k}(1-\alpha_j)\right)\alpha_t Z_f(y_{t+1},x_{t+1})\right\|_\star\right]}_{:=\text{⑦}} + \underbrace{\mathrm{E}\left[\left\|\sum_{t=0}^{k}\left(\prod_{j=t+1}^{k}(1-\alpha_j)\right)\alpha_t \varepsilon_{t+1}^g\right\|_\star\right]}_{:=\text{⑧}}.
$$

Next, we bound ⑧ by using Assumption 1 and by following the proof arguments for bounding ① in Step 2) of the convergence proof for Theorem 1. Then, we have

$$
\text{⑤} \leq \bar{\rho}\,\mathrm{E}\left[\left\|\sum_{t=0}^{k}\left(\prod_{j=t+1}^{k}(1-\alpha_j)\right)\alpha_t\varepsilon_{t+1}^g\right\|_2\right]
$$

$$
\leq \bar{\rho}\sqrt{\mathrm{E}\left[\left\|\sum_{t=0}^{k}\left(\prod_{j=t+1}^{k}(1-\alpha_j)\right)\alpha_t\varepsilon_{t+1}^g\right\|_2^2\right]}
$$

$$
\leq \bar{\rho}\sigma_g\sqrt{\sum_{t=0}^{k}\left(\prod_{j=t+1}^{k}(1-\alpha_j)^2\right)\alpha_t^2},
$$

Therefore,

$$
\mathrm{E}\left[\|e_{k+1}\|_\star\right] \leq \prod_{t=0}^{k}(1-\alpha_t)\mathrm{E}\left[\|e_0\|_\star\right] + \text{⑥} + \text{⑦} + \bar{\rho}\sigma_g\sqrt{\sum_{t=0}^{k}\left(\prod_{j=t+1}^{k}(1-\alpha_j)^2\right)\alpha_t^2}.
$$

Next, we bound ⑥ and ⑦:

$$
\text{⑥} \leq \sum_{t=0}^{k}\left(\prod_{j=t+1}^{k}(1-\alpha_j)\right)(1-\alpha_t)\mathrm{E}\left[\|Z_f(x_t,x_{t+1})\|_\star\right];
$$

$$
\text{⑦} \leq \sum_{t=0}^{k}\left(\prod_{j=t+1}^{k}(1-\alpha_j)\right)\alpha_t\mathrm{E}\left[\|Z_f(y_{t+1},x_{t+1})\|_\star\right]
$$

Since $(1-\alpha_t)+\alpha_t\theta_t^2 = \frac{1-\alpha_t}{\alpha_t} = \theta_t$,

$$
\|Z_f(x_t,x_{t+1})\|_\star \leq \frac{1}{2}(M_0+M_1\|\nabla f(x_{t+1})\|_\star)\|x_t-x_{t+1}\|^2
$$

$$
+\frac{1}{3}M_1(L_0+L_1\|\nabla f(x_{t+1})\|_\star)\exp(L_1\|x_t-x_{t+1}\|)\|x_t-x_{t+1}\|^3
$$

$$
\leq \frac{1}{2}(M_0+M_1\|\nabla f(x_{t+1})\|_\star)\eta_t^2 + \frac{1}{3}M_1(L_0+L_1\|\nabla f(x_{t+1})\|_\star)\exp(L_1\eta_t)\eta_t^3
$$

$$
= \left(\frac{M_0}{2}+\frac{M_1}{3}L_0\exp(L_1\eta_t)\right)\eta_t^2
$$

$$
+\left(\frac{M_1}{2}+\frac{M_1}{3}L_1\eta_t\exp(L_1\eta_t)\right)\eta_t^2\|\nabla f(x_{t+1})\|_\star;
$$

and

$$\|Z_f(y_{t+1}, x_{t+1})\|_\star \leq \frac{1}{2}(M_0 + M_1 \|\nabla f(x_{t+1})\|_\star) \|y_{t+1} - x_{t+1}\|^2$$

$$+ \frac{1}{3}M_1(L_0 + L_1 \|\nabla f(x_{t+1})\|_\star) \exp(L_1 \|y_{t+1} - x_{t+1}\|) \|x_t - x_{t+1}\|^3$$

$$\leq \frac{1}{2}(M_0 + M_1 \|\nabla f(x_{t+1})\|_\star)\theta_t^2 \eta_t^2 + \frac{1}{3}M_1(L_0 + L_1 \|\nabla f(x_{t+1})\|_\star) \exp(L_1 \theta_t \eta_t)\theta_t^3 \eta_t^3$$

$$= \left(\frac{M_0}{2} + \frac{M_1}{3} L_0 \exp(L_1 \theta_t \eta_t)\right) \theta_t^2 \eta_t^2$$

$$+ \left(\frac{M_1}{2} + \frac{M_1}{3} L_1 \theta_t \eta_t \exp(L_1 \theta_t \eta_t)\right) \theta_t^2 \eta_t^2 \|\nabla f(x_{t+1})\|_\star,$$

we obtain

$$⑥ + ⑦ \leq \sum_{t=0}^{k} \left(\prod_{j=t+1}^{k}(1 - \alpha_j)\right) \left(\frac{M_0}{2} + \frac{M_1}{3} L_0 \exp\left(\frac{L_1 \eta_t}{\alpha_t}\right)\right) \theta_t \eta_t^2$$

$$+ \sum_{t=0}^{k} \left(\prod_{j=t+1}^{k}(1 - \alpha_j)\right) \left(\frac{M_1}{2} + \frac{M_1}{3} \cdot \frac{L_1 \eta_t}{\alpha_t} \exp\left(\frac{L_1 \eta_t}{\alpha_t}\right)\right) \theta_t \eta_t^2 \|\nabla f(x_{t+1})\|_\star.$$

Therefore,

$$\mathrm{E}\left[\|e_{k+1}\|_\star\right] \leq \prod_{t=0}^{k}(1 - \alpha_t)\mathrm{E}\left[\|e_0\|_\star\right] + \bar{\rho}\sigma_g \sqrt{\sum_{t=0}^{k}\left(\prod_{j=t+1}^{k}(1 - \alpha_j)^2\right)\alpha_t^2}$$

$$+ \sum_{t=0}^{k}\left(\prod_{j=t+1}^{k}(1 - \alpha_j)\right)\left(\frac{M_0}{2} + \frac{M_1}{3}L_0 \exp\left(\frac{L_1 \eta_t}{\alpha_t}\right)\right)\theta_t \eta_t^2$$

$$+ \sum_{t=0}^{k}\left(\prod_{j=t+1}^{k}(1 - \alpha_j)\right)\left(\frac{M_1}{2} + \frac{M_1}{3}\frac{L_1 \eta_t}{\alpha_t}\exp\left(\frac{L_1 \eta_t}{\alpha_t}\right)\right)\theta_t \eta_t^2 \|\nabla f(x_{t+1})\|_\star.$$

If $\eta_k = \eta$ and $\alpha_k = \alpha$, then

$$\mathrm{E}\left[\|e_{k+1}\|_\star\right] \leq (1 - \alpha)^{k+1}\|e_0\|_\star + \left(\frac{M_0}{2} + \frac{M_1}{3}L_0 \exp\left(\frac{L_1 \eta}{\alpha}\right)\right)\sum_{t=0}^{k}(1 - \alpha)^{k-t}\theta\eta^2$$

$$+ \left(\frac{M_1}{2} + \frac{M_1}{3}\frac{L_1 \eta}{\alpha}\exp\left(\frac{L_1 \eta}{\alpha}\right)\right)\sum_{t=0}^{k}(1 - \alpha)^{k-t}\theta\eta^2 \|\nabla f(x_{t+1})\|_\star$$

$$+ \bar{\rho}\sigma_g \sqrt{\sum_{t=0}^{k}(1 - \alpha)^{2(k-t)}\alpha^2}.$$

Next, since

$$\sum_{t=0}^{k-1}(1 - \alpha)^{k-t} \leq \sum_{j=0}^{\infty}(1 - \alpha)^j = \frac{1}{\alpha},$$

and

$$\sum_{t=0}^{k-1}(1 - \alpha)^{2(k-t)}\alpha^2 \leq \alpha^2 \sum_{j=0}^{\infty}((1 - \alpha)^2)^j = \frac{\alpha^2}{1 - (1 - \alpha)^2} = \frac{\alpha}{2 - \alpha} \overset{\alpha \in [0,1]}{\leq} \alpha,$$

we obtain

$$
\begin{aligned}
\mathrm{E}\left[\|e_{k+1}\|_\star\right] &\leq (1-\alpha)^{k+1}\|e_0\|_\star + \left(\frac{M_0}{2} + \frac{M_1}{3}L_0\exp\left(\frac{L_1\eta}{\alpha}\right)\right)\frac{\eta^2}{\alpha^2} + \bar\rho\sqrt{\alpha}\sigma_g \\
&\quad + \left(\frac{M_1}{2} + \frac{M_1}{3}\frac{L_1\eta}{\alpha}\exp\left(\frac{L_1\eta}{\alpha}\right)\right)\sum_{t=0}^{k}(1-\alpha)^{k-t}\theta\eta^2\|\nabla f(x_{t+1})\|_\star.
\end{aligned}
$$

Therefore, recalling $\varphi = \eta(1 - \exp(L_1\eta)L_1\eta/2)$, we have

$$
\begin{aligned}
\sum_{k=0}^{K}\eta\varphi\mathrm{E}\left[\|\nabla f(x_k)\|_\star\right] &\leq \Delta + \frac{L_0}{2}\sum_{k=0}^{K}\exp(L_1\eta)\eta^2 + 2\eta\sum_{k=0}^{K}\mathrm{E}\left[\|e_k\|_\star\right] \\
&\leq \Delta + \frac{L_0}{2}\exp(L_1\eta)\eta^2(K+1) + 2\eta\sum_{k=0}^{K}(1-\alpha)^k\|e_0\|_\star \\
&\quad + \eta\left(M_0 + \frac{2M_1}{3}L_0\exp\left(\frac{L_1\eta}{\alpha}\right)\right)\frac{\eta^2}{\alpha^2}(K+1) \\
&\quad + \eta\left(M_1 + \frac{2M_1}{3}\frac{L_1\eta}{\alpha}\exp\left(\frac{L_1\eta}{\alpha}\right)\right)\sum_{k=0}^{K}\sum_{t=0}^{k-1}(1-\alpha)^{k-t}\theta\eta^2\|\nabla f(x_{t+1})\|_\star \\
&\quad + 2\eta\bar\rho\sqrt{\alpha}\sigma_g(K+1),
\end{aligned}
$$

where $\Delta_0 = \mathrm{E}\left[f(x_0) - f_{\inf}\right]$.

Next, since

$$
\sum_{k=0}^{K}(1-\alpha)^k \leq \frac{1}{\alpha}, \quad \text{and} \quad \sum_{k=0}^{K}\sum_{t=0}^{k-1}(1-\alpha)^{k-t}\|\nabla f(x_{t+1})\|_\star \leq \frac{1}{\alpha}\sum_{k=0}^{K}\|\nabla f(x_k)\|_\star,
$$

we obtain

$$
\begin{aligned}
\sum_{k=0}^{K}\eta\vartheta\mathrm{E}\left[\|\nabla f(x_k)\|_\star\right] &\leq \Delta + \frac{L_0\eta^2}{2}\exp(L_1\eta)(K+1) + 2\frac{\eta}{\alpha}\|e_0\|_\star \\
&\quad + \left(M_0 + \frac{2M_1}{3}L_0\exp\left(\frac{L_1\eta}{\alpha}\right)\right)\frac{\eta^3}{\alpha^2}(K+1) \\
&\quad + \bar\rho\sqrt{\alpha}\sigma_g(K+1),
\end{aligned}
$$

where $\vartheta := \left(1 - \exp(L_1\eta)\frac{L_1\eta}{2} - \left(M_1 + \frac{2M_1}{3}\cdot\frac{L_1\eta}{\alpha}\exp\left(\frac{L_1\eta}{\alpha}\right)\right)\frac{\eta^2}{\alpha^2}\right)$.

If $\eta \leq \min\left\{\frac{1}{3L_1}, \frac{1}{3\sqrt{M_1}}\right\}\alpha$, then $\eta \leq \min\left\{\frac{1}{3L_1}, \frac{1}{3\sqrt{M_1}}\right\}$ and

$$
\begin{aligned}
\frac{\eta}{2}\sum_{k=0}^{K}\mathrm{E}\left[\|\nabla f(x_k)\|_\star\right] &\leq \Delta + (K+1)\frac{L_0}{2}\exp(L_1\eta)\eta^2 + 2\frac{\eta}{\alpha}\|e_0\|_\star + 2\eta\bar\rho\sqrt{\alpha}\sigma_g(K+1) \\
&\quad + \left(M_0 + \frac{2M_1}{3}L_0\exp\left(\frac{L_1\eta}{\alpha}\right)\right)\frac{\eta^3}{\alpha^2}(K+1).
\end{aligned}
$$

Therefore,

$$
\begin{aligned}
\frac{1}{K+1}\sum_{k=0}^{K}\mathrm{E}\left[\|\nabla f(x_k)\|_\star\right] &\leq \frac{2\Delta}{\eta(K+1)} + L_0\exp(L_1\eta)\eta + \frac{4\|e_0\|_\star}{\alpha(K+1)} + 4\bar\rho\sqrt{\alpha}\sigma_g \\
&\quad + 2\left(M_0 + \frac{2M_1}{3}L_0\exp\left(\frac{L_1\eta}{\alpha}\right)\right)\frac{\eta^2}{\alpha^2}.
\end{aligned}
$$

If $\eta = \frac{\hat{\eta}}{(K+1)^{5/7}}$ with $\hat{\eta} = \min\left\{\frac{1}{3L_1}, \frac{1}{3\sqrt{M_1}}\right\}$ and $\alpha = \frac{1}{(K+1)^{4/7}}$, then

$$
\frac{1}{K+1}\sum_{k=0}^{K} \mathrm{E}\left[\|\nabla f(x_k)\|_\star\right] \leq \frac{2\Delta}{\hat{\eta}(K+1)^{2/7}} + \frac{\hat{\eta}L_0\exp(L_1\hat{\eta})}{(K+1)^{5/7}} + \frac{4\|e_0\|_\star}{(K+1)^{2/7}} + 4\frac{\bar{\rho}\sigma_g}{(K+1)^{2/7}}
$$

$$
+ 2\left(M_0 + \frac{2M_1}{3}L_0\exp(L_1\hat{\eta})\right)\frac{\hat{\eta}^2}{(K+1)^{2/7}}.
$$

Finally, by the fact that $\min_{k\in\{0,1,\ldots,K\}} \mathrm{E}\left[\|\nabla f(x_k)\|_\star\right] \leq \frac{1}{K+1}\sum_{k=0}^{K}\mathrm{E}\left[\|\nabla f(x_k)\|_\star\right]$, we obtain the final result.

## F EXPERIMENTS

To ensure a fair and reproducible comparison, the initialization and hyperparameter schedules were standardized across all experiments.

### F.1 INITIALIZATION

For each experiment, a single initial parameter vector, $x_0$, was generated by drawing from a normal (Gaussian) distribution, $x_0 \sim \mathcal{N}(0, 1)$. The random seed was fixed to ensure that this exact same starting point was used for every algorithm evaluated in that experiment. This "far start" initialization is designed to test the robustness and convergence capabilities of the optimizers from a non-trivial region of the parameter space.

### F.2 STEPSIZE AND MOMENTUM SCHEDULES

The learning rate $\eta_k$ and the momentum parameter $\alpha_k$ are decayed at each iteration $k$. The schedules are chosen based on the theoretical underpinnings of each class of algorithm. Let $\eta_0$ be a pre-defined initial learning rate.

- **For Polyak Momentum:** This first-order method uses its standard theoretically-backed schedule:

$$\alpha_k = \frac{1}{\sqrt{k+1}}, \quad \eta_k = \frac{\eta_0}{(k+1)^{3/4}}.$$

- **For Extrapolated Momentum:** This method uses a distinct schedule designed for its update rule:

$$\alpha_k = \frac{1}{(k+1)^{4/7}}, \quad \eta_k = \frac{\eta_0}{(k+1)^{5/7}}.$$

- **For all Second-Order Momentum Variants (SOM-V1, SOM-V2, $\beta$-SOM-V1, and $\beta$-SOM-V2):** These methods share a common schedule for their primary learning rate and momentum parameters:

$$\alpha_k = \frac{1}{(k+1)^{2/3}}, \quad \eta_k = \frac{\eta_0}{(k+1)^{2/3}}.$$

These schedules ensure that the step sizes and momentum contributions diminish over time, a necessary condition for convergence in stochastic optimization.

### F.3 NONCONVEX LOGISTIC REGRESSION

We evaluate the performance of several stochastic optimization algorithms on a composite nonconvex problem. The objective is to benchmark their convergence speed and stability on a logistic regression task augmented with a non-convex regularizer.

The model is a standard logistic regression classifier. The experiment is conducted on the `splice` dataset from the libsvm library, which contains 1000 training samples and 60 features.

The optimization objective is to minimize a composite function $f(x)$, defined as:

$$\min_{x \in \mathbb{R}^d} f(x) = \mathcal{L}(x) + R(x)$$

where $x \in \mathbb{R}^{60}$ is the vector of model parameters. The loss term $\mathcal{L}(x)$ is the standard mean logistic loss:

$$\mathcal{L}(x) = \frac{1}{N} \sum_{i=1}^{N} \log(1 + \exp(-y_i a_i^T x))$$

The regularization term $R(x)$ is the non-convex Welsch regularizer, defined as:

$$R(x) = \lambda \sum_{j=1}^{d} \frac{x_j^2}{1 + x_j^2}$$

The regularization hyperparameter $\lambda$ is set to $0.01$.

Figure 5 displays the convergence behavior of the three algorithms over 20000 iterations. The training loss plot shows that the second-order methods containing Hessian information (Variant 1, Variant 2) outperform the first-order Polyak Momentum. Among these, Variant 2 achieves the fastest convergence and reaches the lowest final loss value.

The gradient norm plot corroborates these findings. While all methods exhibit stochastic oscillations, the overall trend for the second-order methods is a more rapid and consistent decrease in the gradient norm. The Polyak Momentum method converges to a region with a substantially higher gradient norm, indicating a less optimal solution.

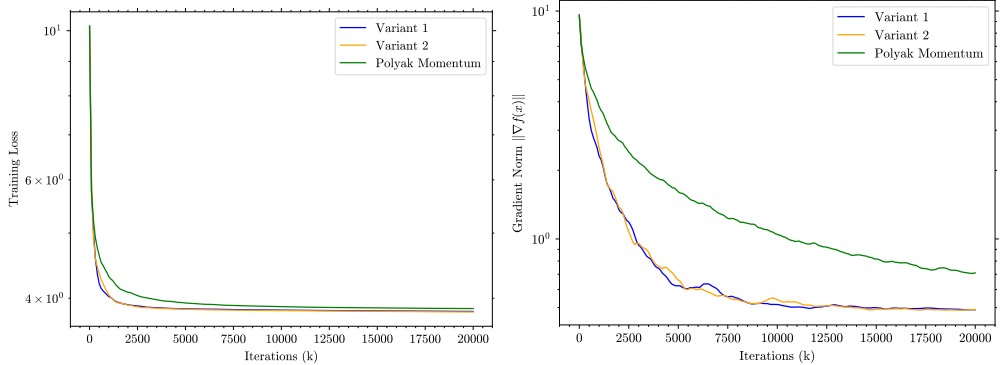

Figure 5: Main algorithm comparison for logistic regression. Training loss vs. iterations. Gradient norm vs. iterations.

### F.4 Multi-Layer Perceptron

We investigate the performance of several stochastic optimization algorithms on a non-convex binary classification problem. Our goal is to compare their convergence properties.

The model is a Multi-Layer Perceptron (MLP) with two hidden layers, implemented in PyTorch. The experiment is conducted on the `splice` dataset, obtained from the libsvm library. It consists of 1000 training samples, each with 60 features.

The optimization objective is to minimize a composite function $f(x)$, which includes a standard loss term and a non-convex regularizer:

$$\min_{x \in \mathbb{R}^d} f(x) = \mathcal{L}(x) + R(x)$$

where $x$ represents the flattened vector of all model parameters.

The loss term, $\mathcal{L}(x)$, is the mean Binary Cross-Entropy with Logits loss, calculated over the entire training dataset.

The regularization term, $R(x)$, is the non-convex Welsch regularizer, chosen to create a more challenging optimization landscape. It is defined as:

$$R(x) = \lambda \sum_{i=1}^{d} \frac{x_i^2}{1 + x_i^2}$$

where $\lambda$ is the regularization hyperparameter, set to $0.01$.

We set the theoretical learning rates for the algorithms considered.

The results are presented in two separate comparisons. Figure 6 compares four momentum-based algorithms. Among these, SOM-V2 achieves the lowest final training loss. Extrapolated Momentum and SOM-V1 perform similarly, while Polyak Momentum converges to a slightly higher loss. The gradient norm plot for this group shows that all methods successfully reduce the gradient's magnitude, though their convergence paths exhibit the high variance characteristic of stochastic methods.

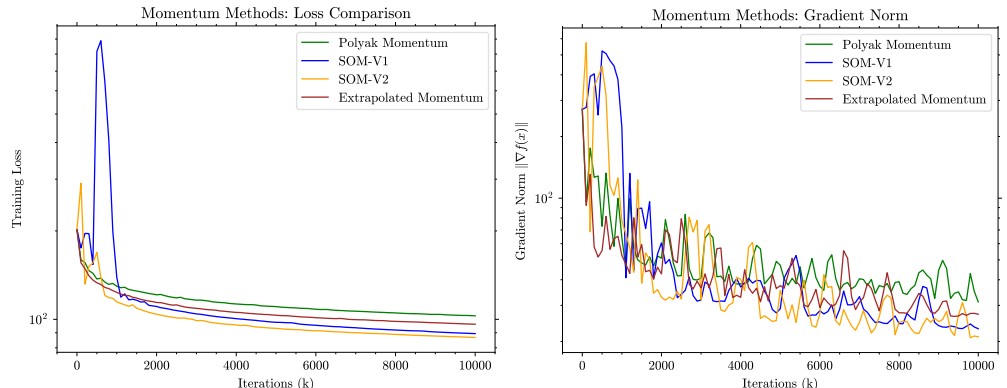

Figure 6: Comparison of momentum-based methods. Left: Training loss vs. iterations. Right: Gradient norm vs. iterations.

Figure 7 compares the standard second-order momentum methods (SOM-V1 and SOM-V2) with their $\beta$-HVP counterparts. The results clearly show the benefit of the additional Hessian term, as both $\beta$-SOM-V1 and $\beta$-SOM-V2 outperform their respective base variants. $\beta$-SOM-V2 demonstrates the strongest overall performance, converging to the lowest training loss of all tested algorithms. The gradient norm plot also suggests that the $\beta$-SOM methods, particularly $\beta$-SOM-V2, offer a more consistent decrease in gradient magnitude compared to the standard SOM-V1 and SOM-V2 methods.

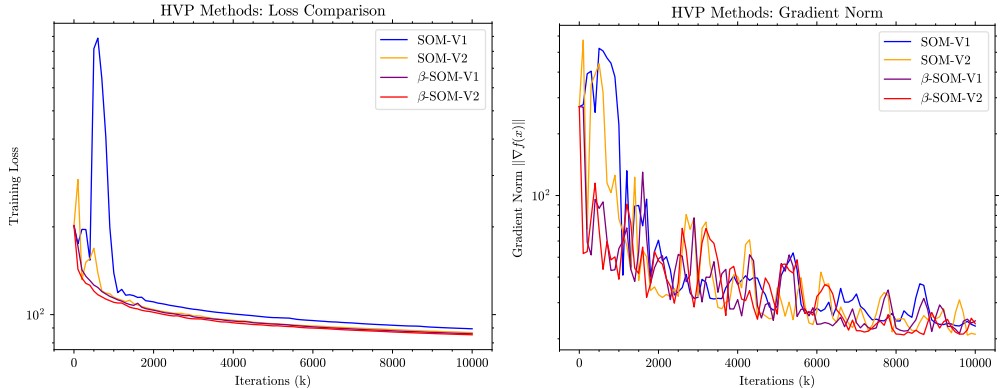

Figure 7: Comparison of HVP-based methods. Left: Training loss vs. iterations. Right: Gradient norm vs. iterations.

## F.5 RECURRENT NEURAL NETWORK TRAINING

The model is a two-layer LSTM network designed for word-level language modeling. The architecture consists of an embedding layer (200 dimensions), a two-layer LSTM core with hidden units of size 200, and a final linear decoder layer to produce logits over the vocabulary.

The experiments are conducted on the Penn Treebank (PTB) dataset, a standard and widely-used benchmark for evaluating language models.

Key statistics and preprocessing steps are as follows:

- **Corpus Size:** The dataset is split into training, validation, and testing sets. The training portion, used in our experiment, contains approximately $929,000$ tokens.
- **Vocabulary:** A dictionary is constructed from the unique words present in the training data. An end-of-sentence token, <eos>, is added to each sentence, resulting in a total vocabulary size of $10,000$ unique tokens.

- **Tokenization:** The raw text is tokenized by splitting on whitespace, and each word is converted into its corresponding integer index from the vocabulary.

- **Batching:** For training, the entire sequence of token IDs is reshaped into a fixed number of parallel streams (a batch size of 20 in our case). The model is then trained on sequential chunks of this data using Truncated Backpropagation Through Time (BPTT) with a sequence length of 35.

The objective is to minimize the standard Cross-Entropy Loss. The parameter update follows the LMO-based rule:

$$x_{k+1} = x_k + \text{lmo}(m_k)$$

where the Linear Minimization Oracle, $\text{lmo}(m_k)$, is defined as:

$$\text{lmo}(m_k) := \arg\min_{\|v\| \le \eta_k} \langle m_k, v \rangle.$$

We compare six algorithms: Polyak Momentum, SOM-V1, SOM-V2, our proposed $\beta$-SOM-V1 and $\beta$-SOM-V2 variants, and Extrapolated Momentum. The hyperparameter schedules for $\alpha_k$ and $\eta_k$ are set to their theoretical values for each respective algorithm.

### F.6 THE CASE WHEN $\|\cdot\|$ IS A EUCLIDEAN NORM

For this experiment, we use the Euclidean ($\ell_2$) norm. The results of the LMO-based training, presented in Figure 8, reveal a significant divergence in performance between the first-order and more advanced momentum methods. The simple momentum update rule is less efficient at navigating

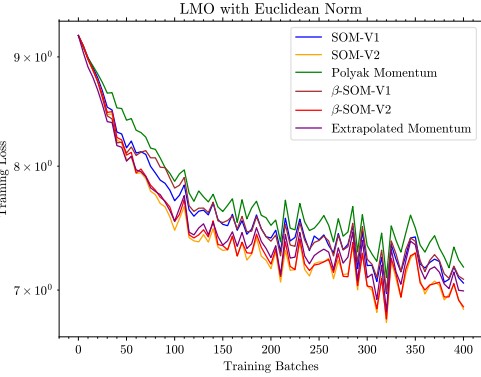

Figure 8: Training loss vs. training batches for the LSTM model using an LMO update with the Euclidean norm. The plot highlights the stronger performance of the other five methods over Polyak Momentum.

the loss landscape for this task. In contrast, the other five algorithms, which all incorporate either second-order (HVP) or extrapolated gradient information, form a tight cluster of high-performing methods. They converge faster and to a lower loss value than Polyak Momentum. Within this cluster, SOM-V2 and $\beta$-SOM-V2 often achieve a marginally lower loss, suggesting that calculating the curvature information at the current point $x_k$ (rather than the interpolated point used by SOM-V1 and $\beta$-SOM-V1) may offer a slight advantage. The strong performance of these five methods indicates that the LMO update rule is a highly effective stabilization technique when paired with momentum strategies that utilize more sophisticated directional information.

### F.7 THE CASE WHEN $\|\cdot\|$ IS $\ell_\infty$-NORM

To further investigate the effect of the update geometry, we conducted a parallel experiment where the LMO is constrained by the Infinity norm ($\ell_\infty$). The resulting training loss curves are presented in Figure 9. The most immediate observation is that while all algorithms still successfully guide the training process toward a lower loss, the convergence path is significantly less stable. The loss curves for all methods are characterized by high-frequency, large-amplitude oscillations.

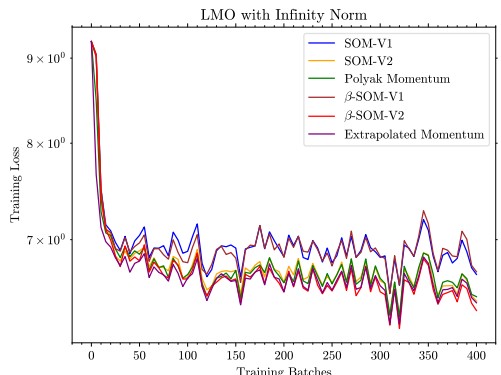

Figure 9: Training loss vs. training batches for the LSTM model using an LMO update with the Infinity norm. The optimization process is successful but visibly less stable than with the Euclidean norm.

This instability is an expected consequence of the $\ell_\infty$ LMO. Unlike the smooth, normalized direction vector produced by the $\ell_2$ norm, the $\ell_\infty$ LMO generates a more aggressive update where every component of the step vector is pushed to its maximum value. This results in a more chaotic exploration of the parameter space, leading to the observed volatility in the training loss.

An important consequence of this instability is that the clear performance hierarchy observed in the Euclidean experiment has vanished. The loss curves for all six algorithms are tightly intertwined, and no single method demonstrates a consistent advantage. This suggests that the high variance and non-smooth nature of the $\ell_\infty$ update step dominate the more subtle directional corrections offered by the second-order and extrapolated momentum terms.

While the final loss values appear comparable to those achieved with the $\ell_2$ norm, the convergence path is significantly less stable, making the Euclidean norm the more reliable and predictable choice for this particular language modeling task.

## LLM Use Acknowledgment

In this paper, we used large language models (LLMs) to assist with grammar and wording during the preparation of the manuscript. We did not use LLMs to derive convergence theorems, generate empirical plots, or search for citations. This usage is in accordance with two primary LLM-related policies.

