# OpenReview forum: "Better LMO-based Momentum Methods with Second-Order Information"
_ICLR.cc/2026/Conference — Submitted to ICLR 2026_

### Official Review · Reviewer_8Ycx · 2025-10-29

**Soundness:** 3
**Presentation:** 4
**Contribution:** 3
**Rating:** 6
**Confidence:** 1

**Summary:**

The paper integrates second-order information, via Hessian-Corrected Momentum (HCM), into the LMO-based momentum framework, and proposes a rigorous theoretical analysis proving that the second-order LMO methods achieve an accelerated convergence rate of  $\mathcal{O}(1/K^{1/3})$, under a relaxed $(L_0, L_1)$-smoothness assumption and in an arbitrary norm setting. The authors support their theoretical findings with experiments on several non-convex problems.

**Strengths:**

1. The paper is well-written and structured, providing a clear summary of the previous methods. Moreover, this paper clearly introduces the existing theoretical challenges, which are mathematically integrating second-order Hessian information into the arbitrary-norm LMO framework and proving its fast $\mathcal{O}(1/K^{1/3})$ rate.

2. The paper proves second-order methods like HCM achieve an accelerated $\mathcal{O}(1/K^{1/3})$ rate, matching the optimal known rate for this problem class and breaking the $\mathcal{O}(1/K^{1/4})$ barrier of their first-order counterparts. The theoretical contribution is non-trivial, because the proofs hold under relaxed smoothness assumption in arbitrary norms settings.

3. Experiments in various tasks validates the theoretical claims in this paper.

**Weaknesses:**

1. While the paper's motivation mentions optimizers like Muon, which are utilized in large-scale models, the empirical validation in this paper is conducted on relatively small-scale problems.

2. The paper's theoretical guarantees are general, holding for arbitrary norms. However, the experimental validations seem limited to the $l_2$ and $l_\infty$ cases. Moreover, the $l_\infty$ case seems to show only minor benefit (see Q1).

**Questions:**

1. In Appendix F.7, the paper provides the analysis where the LMO is constrained by the Infinity norm. Is your experiment here showing that the theoretical acceleration does not translate into a observable speed-up on $l_\infty$? If the theoretical acceleration disappears in the $l_\infty$ setting due to instability, what is the practical benefit of the arbitrary norm guarantee? I feel like this is a disconnect between the theory and the experimental validation.

---

> ### Author Response · Authors · 2025-11-21
>
> We sincerely thank you for recognizing the significance of extending the optimal $\mathcal{O}(K^{-1/3})$ convergence rate to the arbitrary norm LMO framework under generalized smoothness.
>
> We would like to address your concerns below.
>
>
> **Disconnect: $\ell_\infty$ Instability vs. Theoretical Acceleration:** We would like to clarify that the conclusion that our algorithms are unstable under the $\ell_\infty$-norm is misleading. Our second variant of second-order momentum (V2) still consistently outperforms the LMO-based algorithms using Polyak momentum and extrapolated momentum under the $\ell_\infty$-norm. The observed instability only pertains to our first variant of second-order momentum (V1). This divergence is likely due to the inherent geometry of the LMO step under the $\ell_\infty$-norm. Unlike the smooth, isotropic $\ell_2$-norm, the LMO for the $\ell_\infty$-norm generates a highly aggressive, boundary-hitting step (a sign vector). This aggressive update introduces distinct non-smoothness that clashes with the specific curvature correction mechanism in Variant 1, leading to unstable trajectories for that specific variant. Variant 2's superior performance confirms that the theoretical acceleration translates, but V1's failure serves as a useful diagnostic for understanding how different correction mechanisms interact with sharp, non-Euclidean geometry.
>
>
> **Empirical validation against Muon/Scion under matrix norms:**   We agree that conducting additional experiments on learning problems where the norm is a matrix norm (e.g., Nuclear Norm), which requires the LMO framework, will significantly strengthen our empirical demonstration of the algorithms' applicability. We anticipate that our proposed algorithms, which incorporate second-order momentum (Hessian-corrected), will substantially outperform first-order LMO baselines like Muon and Scion, which rely only on Polyak momentum. This claim is strongly supported by our current results, which consistently show that our second-order LMO algorithms yield faster convergence than their Polyak momentum counterparts across both Euclidean ($\ell_2$) and non-Euclidean ($\ell_\infty$) norm settings.

---

> > ### Comment · Reviewer_8Ycx · 2025-11-25
> >
> > Thank you for your clarification, and I will keep my score.

---

### Official Review · Reviewer_jQAq · 2025-10-30

**Soundness:** 4
**Presentation:** 4
**Contribution:** 4
**Rating:** 8
**Confidence:** 2

**Summary:**

The paper theoretically provesthat the proposed LMO-HCM methods  can achieve a convergence rate of $O(1/K^{1/3})$.
This rate is superior to the $O(1/K^{1/4})$ of standard LMO momentum and matches the optimal known rate for non-convex stochastic optimization.
Most critically, this $O(1/K^{1/3})$ rate is guaranteed under both **"relaxed smoothness" ($(L_0, L_1)$-smoothness)** and an **"arbitrary norm"** setting simultaneously, greatly expanding the applicability of second-order momentum methods in deep learning theory.
In training experiments on MLP and LSTM models, the authors demonstrate that their methodssignificantly outperform standard Polyak momentum and Extrapolated momentum in reducing training loss and gradient norm .

**Strengths:**

The core contribution is extending the optimal $O(1/K^{1/3})$ convergence rate from Euclidean settings to the arbitrary norm LMO framework, crucially under the more practical $(L_0, L_1)$ relaxed smoothness assumption.

On non-convex tasks like MLPs and LSTMs, the proposed SOM-V2 and $\beta$-SOM-V2 methods consistently and significantly outperform the Polyak momentum baseline in both convergence speed and final loss, validating the theory.

The method naturally applies to various norms LMO can handle. Experiments demonstrate this flexibility by comparing performance under both $l_2$ and $l_{\infty}$ norms.

**Weaknesses:**

This paper does not provide result about matrix norms, it is good to see it but the current version is already good enough.

**Questions:**

The experiments compare iteration counts, but Algorithm 1 introduces a Hessian-vector product (HVP) computation at every step. In practical training (e.g., for LSTM), how does this extra computational overhead (wall-clock time) from HVP compare to first-order methods (like Polyak Momentum)?

The core of the paper is the LMO framework (applicable to arbitrary norms), but the experiments mainly focus on the $l_2$ and $l_{\infty}$ norms (Figs 8, 9). How does the method perform under more complex norms that truly require an LMO, such as the matrix norms used in Muon/Scion?

---

> ### Author Response · Authors · 2025-11-21
>
> We sincerely thank you for recognizing the significance of extending the optimal $\mathcal{O}(K^{-1/3})$ convergence rate to the arbitrary norm LMO framework under generalized smoothness.
> We would like to address the questions regarding complexity of computing Hessian-vector products and other norms below.
>
> **Computational complexity of Hessian-vector products:** The Hessian-vector product, $\nabla^2 f(x) v$ for arbitrary $x,v \in \mathbb{R}^d$, can be computed efficiently without evaluating explicit Hessians. By utilizing the automatic differentiation package in Pytorch [1], the product is evaluated via a double backward pass using the identity:
>
> $$
> \nabla^2 f(x) v = \nabla_x \langle \nabla_x f(x), v \rangle.
> $$
>
> Consequently, the computational complexity is $\mathcal{O}(d)$ where $d$ is the number of model parameters, scaling linearly. In practice, the cost of this operation is approximately equivalent to 2-3 gradient evaluations, ensuring the method remains scalable for high-dimensional models.
>
> [1] Paszke, Adam, et al. "Automatic differentiation in pytorch." (2017).
>
> **Convergence with respect to the training time:**  Total training time is the product of the per-iteration cost and the iteration complexity. While the HVP introduces a constant-factor overhead (2-3x higher than first-order baselines), the second-order momentum improves the convergence rate, significantly reducing the total number of iterations required. This speedup often outweighs the HVP overhead. We will include a dedicated Loss vs. Wall-Clock Time analysis in the revised manuscript to demonstrate this practical benefit.
>
> **Convergence against Muon/Scion:** We agree that conducting additional experiments under matrix norms will significantly strengthen our empirical contributions. We anticipate that our proposed algorithms, which incorporate second-order momentum (Hessian-corrected), will substantially outperform Muon and Scion algorithms, which rely only on first-order Polyak momentum. This claim is strongly supported by our current results, which consistently show that our second-order LMO algorithms yield faster convergence than their Polyak momentum counterparts across both Euclidean ($\ell_2$) and non-Euclidean ($\ell_\infty$) norm settings.

---

### Official Review · Reviewer_FMro · 2025-11-01

**Soundness:** 3
**Presentation:** 3
**Contribution:** 3
**Rating:** 6
**Confidence:** 2

**Summary:**

The paper studies momentum methods within the Linear Minimization Oracle (LMO) framework and integrates second-order (Hessian-corrected) momentum to obtain faster convergence under relaxed smoothness and arbitrary norm settings. Concretely, it adapts two known second-order momentum variants to LMO (Algorithm 1) and proves $O(K^{-1/3})$ rates in expected gradient norm, improving upon the $O(K^{-1/4})$ guarantees for LMO methods with Polyak momentum and matching best-known rates for second-order momentum in Euclidean $L$-smooth settings (Theorems 1-2). The analysis relies on symmetric $(L_0,L_1)$ gradient smoothness and, for one variant, symmetric $(M_0,M_1)$ Hessian smoothness, with unbiased, variance-bounded gradient/Hessian oracles. Empirically, on MLP and LSTM training (plus a logistic regression task in the appendix), the proposed second-order LMO methods (with and without a scaling factor $\beta_k$ outperform Polyak and extrapolated momentum in training loss and gradient norm; $\ell_\infty$ geometry yields less stable trajectories than $\ell_2$.

**Strengths:**

1. Establishes $O(K^{-1/3})$ convergence for LMO-based momentum with arbitrary norms and relaxed smoothness, improving on the $O(K^{-1/4})$ bound for LMO+Polyak momentum and aligning with best-known second-order momentum rates in Euclidean settings (Theorems 1-2).
2. Well-structured presentation: Algorithm 1 is easy to implement, assumptions are grouped and referenced, and Table 1 positions the results against prior LMO and second-order momentum rates; figures make the geometry ($\ell_2$ vs $\ell_\infty$) effects concrete.
3. Numerical Experiments: On MLP and LSTM tasks, the proposed second-order LMO variants (including the $\beta_k$-scaled version) consistently reduce training loss and gradient norm faster than Polyak and extrapolated momentum, and the instability under $\ell_\infty$ geometry is surfaced as a useful diagnostic.

**Weaknesses:**

1. Empirical validation is modest in scope and scale: MLP on a 1k-sample dataset and a PTB LSTM, with plots primarily of training loss and gradient norm; there are no validation/test metrics (e.g., perplexity), runtime, or wall-clock/throughput comparisons to quantify the extra cost of Hessian-vector products.

2. The paper cites that HVPs are “roughly the same time as computing the gradient,” but does not measure this in practice; thus the compute-efficiency trade-off of the proposed methods remains unclear.

3. Finally, while related work covers STORM/MARS and other improved momentum variants, the empirical comparison omits them, limiting the practical positioning of the proposed methods relative to the strongest baselines.

**Questions:**

See weaknesses. In particular:

1. Can you expand the empirical validation beyond an MLP on a 1k-sample dataset and a PTB LSTM?

2. Can you provide measured timings (per step/epoch) to clarify the trade-off between HVP vs computing the gradient?

3. Could you add baselines such as STORM and MARS (and other improved momentum variants) to better compare the proposed methods?

---

> ### Author Response · Authors · 2025-11-21
>
> We apprecite your comments, which are very helpful for us to improve the quality of our paper. We address your concerns below.
>
> **Scope of experiments:** Our experiments on MLP and LSTM models are designed to validate the applicability of our proposed algorithms under different norms. Specifically, our algorithms using the second-order momentum outperform existing LMO-based algorithms using the Polyak momentum and the extrapolated momentum. We reported the training loss and gradient norm, because they are direct proxies for algorithmic convergence, which are central to our convergence theorems. While validation metrics (like perplexity) are crucial for generalization, they can be confounded by regularization and architecture. However, we agree that additional experiments on larger models, where the norm is a matrix norm, will strengthen our empirical contributions to demonstrate the applicability of our proposed algorithms.
>
> **Computational complexity of Hessian-vector products:** The Hessian-vector product, $\nabla^2 f(x) v$ for arbitrary $x,v \in \mathbb{R}^d$, can be computed efficiently without evaluating explicit Hessians. By utilizing the automatic differentiation package in Pytorch [1], the product is evaluated via a double backward pass using the identity:
>
> $$
> \nabla^2 f(x) v = \nabla_x \langle \nabla_x f(x), v \rangle.
> $$
>
> Consequently, the computational complexity is $\mathcal{O}(d)$ where $d$ is the number of model parameters, scaling linearly. In practice, the cost of this operation is approximately equivalent to 2-3 gradient evaluations, ensuring the method remains scalable for high-dimensional models.
>
> [1] Paszke, Adam, et al. "Automatic differentiation in pytorch." (2017).
>
> **Convergence with respect to the training time:** Total training time is the product between per-iteration cost and iteration complexity. While the Hessian-vector product introduces a constant-factor overhead per iteration (roughly 2-3 times higher than the first-order baselines), second-order momentum improves the convergence rate, thus reducing the total number of iteration required to reach target accuracy. In the revised manuscript, we will add the discussion on this trade-off.
>
> **Empirical convergence against first-order baselines (STORM, MARS):** Our LMO-based algorithms with second-order momentum achieve the convergence rate of $\mathcal{O}(1/K^{1/3})$ in the expected gradient norm. This matches existing convergence guarantees of other variance-reduced momentum algorithms, such as STORM and MARS. Given identical theoretical stepsizes, we expect our algorithms to yield comparable convergence performance.

---

### Official Review · Reviewer_J27p · 2025-11-01

**Soundness:** 2
**Presentation:** 3
**Contribution:** 2
**Rating:** 4
**Confidence:** 3

**Summary:**

This paper studies LMO-type optimizer combined with second-order momentum. LMO optimizers is a generalization of the recently popular Muon optimizer, which has update form $x_t = x_{t-1} - \eta_t lmo(m_t)$, where $m_t$ is the momentum buffer and $lmo(g) = \mathrm{argmin}\_{\\|x\\|\le 1} \langle g,x\rangle$ where $\\|\cdot\\|$ could be an arbitrary non-Euclidean norm. Unlike standard previous analysis of Muon and general LMO optimizers which used standard first-order momentum $m_t = \beta_t m_{t-1} + (1-\beta_t) g_t$, this paper considers Hessian-corrected momentum with an additional Hessian-vector product $\nabla^2 f_t(x_t) (x_t-x_{t-1})$. With the additional second-order information and second-order smoothness assumption, this paper shows that the resulting optimizer achieves $O(1/K^{1/3})$ convergence rate, matching the optimal rate using Euclidean norm. Finally, this paper also includes experiment results comparing the proposed algorithm with previous ones.

**Strengths:**

This paper extends the study of LMO-type optimizers by incorporating second-order Hessian-corrected momentum. It provides convergence analysis and proves that LMO-type optimizer with second-order momentum achieves the optimal $O(1/K^{1/3})$ rate under second-order smoothness. Moreover, empirical experiments also show that the proposed optimizer has better performance compared to other baselines.

**Weaknesses:**

There are two main concerns overall:

- I find the discussion related to variance reduction algorithms and the Mars scaling factor confusing and deviated from the main part of this paper. From my understanding, variance reduction algorithms are vastly different from Hessian-corrected momentum. While this paper seems to focus on the latter, the former seems unrelated. Moreover, it is unclear to me what's the role of the scaling factor $\beta_t / (1-\alpha_t)$ in the proposed Algorithm 1 and related discussion is absent. More importantly, in the convergence analysis (Thm 1 and 2), $\beta_t$ is set of $1-\alpha_t$, making the scaling factor constantly one. It seems to me that adding this scaling factor is totally unnecessary, since it does not give any advantage and is cancelled anyways.

- The convergence rate has constants $\bar \rho, \underline \rho, \bar \theta, \underline\theta $ that are carried over from the variance bound with respect to Euclidean norm. These constants could pick up implicit dimension dependence and significantly weakens the adaptivity to geometry using non-Euclidean arbitrary norms. For example, consider the infinity norm $\\|\cdot\\|_\infty$ and its dual one-norm, which has $\underline\theta = 1/\sqrt{d}, \bar\theta=1$ and $\bar\rho=\sqrt{d}, \underline\rho=1$.
With this limitation, the convergence rate does not improve from previous rates of Hessian-corrected momentum with Euclidean norm (e.g., Salehkaleybar et al. 2022 and Tran & Cutkosky 2022). In fact, we could simply apply the identity $\underline\rho \\|\cdot\\|\_2 \le \\|\cdot\\|\_* \le \bar\rho \\|\cdot\\|\_2$ on the previous Euclidean norm bound of form $\mathbb{E} \\| \nabla F(w)\\|_2 = O(1/K^{1/3})$ (e.g., Tran & Cutkosky 2022 Thm 1) and get the same result.

**Questions:**

- A few comments on the related works:
  - line 65 typo: $K^{2/5}$ -> $K^{2/7}$; also I think [1] should be a better reference for this rate, as the current reference Cutkosky & Mehta 2021 focuses more on heavy-tail noise and the core technique follows from [1].
  - For the lower bound part, Arjevani et al 2023 proves the lower bound of standard smooth non-convex optimization is $O(1/K^{1/4})$ and lower bound of mean-square smooth (which corresponds to variance reduction algorithms such as Storm and Mars) is $O(1/K^{1/3})$. However, such lower bounds are only for first-order oracles and doesn't apply to results like Salehkaleybar et al 2022 and Tran & Cutkosky 2022. Instead, the lower bound for second-order smooth problems with second-order oracles is from another paper [2], which proves a corresponding lower bound $O(1/K^{1/3})$.
  - line 220: I think [3] should also be mentioned as it's the original source of signSGD.

- Assumption 4: typo: $\\|\nabla^2\\|_ *$ -> $\\|\nabla^2\\|_{op}$

- The empirical experiments only include simple tasks with small scale models such as MLP and two-layer LSTM. Larger scale models on more complicated tasks are encouraged to better demonstrate the performance of the proposed optimizer.

[1] Cutkosky, A. and Mehta, H., “Momentum Improves Normalized SGD”

[2] Arjevani, Y., Carmon, Y., Duchi, J. C., Foster, D. J., Sekhari, A., and Sridharan, K., “Second-Order Information in Non-Convex Stochastic Optimization: Power and Limitations”

[3] Bernstein, J., Wang, Y.-X., Azizzadenesheli, K., and Anandkumar, A., “signSGD: Compressed Optimisation for Non-Convex Problems”

---

> ### Author Response · Authors · 2025-11-21
>
> **Concern 1: Hessian correction and the need of scaling factors**
>
> **Hessian correction:** Although Hessian-corrected and variance-reduced momentum appear distinct at the first glance, both share the goal of creating a low-variance gradient estimator for the momentum update. The term containing the Hessian-vector product $\nabla^2 f(x)(x_{k+1}-x_k)$ serves a Hessian correction alternative to the gradient difference used in STORM/MARS $\nabla f(x_{k+1}) - \nabla f(x_k)$.
>
> The equivalence between our algorithms and MARS can be proved by Taylor’s series expansion::
> $$
> \nabla f(x_{k+1}) - \nabla f(x_k) \approx \nabla^2 f(x_{k+1})(x_{k+1}-x_k).
> $$
> Specifically, while MARS updates the momentum $m_k$ via the gradient difference:
> $$
> m_{k+1} = (1-\alpha_k)\left(m_k + \frac{\beta_k}{1-\alpha_k}[ \nabla f_{\xi_{k+1}}(x_{k+1}) - \nabla f_{\xi_{k+1}}(x_{k})   ]  \right) + \alpha_k \nabla f_{\xi_{k+1}}(x_k),
> $$
> our proposed algorithms substitutes this difference with the Hessian-corrected term:
> $$
> m_{k+1} = (1-\alpha_k)\left(m_k + \frac{\beta_k}{1-\alpha_k}[ \nabla^2 f_{\xi_{k+1}}(x_{k+1})(x_{k+1} - x_k)  ]  \right) + \alpha_k \nabla f_{\xi_{k+1}}(x_k).
> $$
> Inspired by this equivalence, our proposed algorithms using the Hessian correction are theoretically shown to enjoy the same $\mathcal{O}(1/K^{1/3})$ convergence rate as STORM/MARS.
>
> **Justification for $\beta_k/(1-\alpha_k)$:** We include $\beta_k/(1-\alpha_k)$ to provide the unified description of our proposed algorithms. Our update rules are inspired by  the MARS algorithms, which incorporate the $\beta_k$ factor to improve the performance over MVR algorithms. While our analysis focuses on the case when $\beta_k=1-\alpha_k$, our theory can be extended for general $\beta_k$ values. Consequently, the resulting convergence bound with any $\beta_k$ values will exhibit one additional error term resulting from the momentum update:
> $$
> e_{k+1} = (1-\alpha_k)e_k  + (1-\alpha_k)W_{k+1} + \alpha_k V_{k+1} + (1-\alpha_k)(1-\beta_k/(1-\alpha_k))\nabla^2 f_{\xi_{k+1}}(x_{k+1} - x_k),
> $$
> where $e_k=\nabla f(x_k)-m_k$ is the momentum error term. Note that the final error term is eliminated when $\beta_k=1-\alpha_k$ that our theory considers. We verify the benefit of $\beta_k/(1-\alpha_k)$ on our proposed algorithms in Figure 1.
>
>
> **Concern 2 on the results**
>
> We kindly disagree with you. As Reviewers jQAq and 6CUC point out, we prove the $\mathcal{O}(1/K^{1/3})$ convergence under both relaxed smoothness, e.g. $(L_0,L_1)$-smoothness and $(M_0, M_1)$-smoothness, and arbitrary norm settings. Our results extend beyond traditional smoothness and Euclidean settings, under which prior works (e.g., Salehkaleybar et al. 2022 andTran & Cutkosky 2022) analyzed the convergence of second-order momentum algorithms. Therefore, our results are more general than, and our proofs are more challenging than existing literature on second-order momentum and variance-reduced momentum algorithms consider (traditional smoothness and Euclidean norm settings).
>
>
>
> > **line 65 typo:  K^{2/5} -> K^{2/7} ; also I think [1] should be a better reference for this rate, as the current reference Cutkosky & Mehta 2021 focuses more on heavy-tail noise and the core technique follows from [1].**
>
> Thank you for pointing out this typo. We fix this typo in the revised manuscript.
>
> > **... the lower bound for second-order smooth problems with second-order oracles is from another paper [2], which proves a corresponding lower bound $\mathcal{O}(1/K^{1/3})$.**
>
> Thank you for this suggestion. We cite [2], rather than Arjevani et al. (2023), concerning the lower bound for second-order smooth problems with second-order oracles, which are applicable to our proposed algorithms.
>
> > **line 220: I think [3] should also be mentioned as it's the original source of signSGD.**
>
> We add [3] in the revised version by editing the text on Line 220 as follows: “sign stochastic momentum methods (Jiang et al., 2025), which applies momentum updates to improve the convergence of signSGD by Bernstein et al. (2018) when we let $\| \cdot \| = \| \cdot \|_\infty$.”
>
> > **Assumption 4: typo:  $\| \nabla^2 \|_\star$ -> $\| \nabla^2 \|_{op}$**
>
> We agree. We rewrite the equation shown in Assumption 4 as follows: For $x,y,w\in\mathbb{R}^d$,
> $$
> \|\| (\nabla^2 f(x) - \nabla^2 f(y))w   \|\|_{\star} \leq (M_0 + M_1\sup_{\theta \in [0,1]]}\|\|\nabla f(\theta x + (1-\theta)y)\|\|_{\star})\|\| x-y \|\| \|\| w \|\|.
> $$
>
> **On experiments for larger scale models**
>
> Our experiments on MLP and LSTM validate that our proposed algorithms achieve faster convergence than Polyak and extrapolated momentum under different norm settings, thus demonstrating the practical viability of using our algorithms for problems under various geometry landscapes. However, we agree that additional experiments on larger models, where the norm is a matrix norm, will strengthen our empirical contributions to demonstrate the applicability of our proposed algorithms.

---

### Official Review · Reviewer_6CUC · 2025-11-08

**Soundness:** 3
**Presentation:** 3
**Contribution:** 3
**Rating:** 6
**Confidence:** 3

**Summary:**

The paper introduces two second-order momentum variants for the Linear Minimization Oracle (LMO) framework, using Hessian-vector products as correction terms. The authors prove an improved convergence rate of O(1/K^{1/3}) under relaxed smoothness and arbitrary norm assumptions, extending beyond the usual Euclidean setting. Experiments on MLPs and LSTMs show consistent improvements over standard Polyak and extrapolated momentum, supporting the theoretical results.

**Strengths:**

* Strong theoretical contribution that raises the convergence rate of LMO-based methods from O(1/K^{1/4}) to O(1/K^{1/3}).
* Analysis covers arbitrary norms and relaxed smoothness, making the results broadly applicable to deep learning settings.
* Two well-motivated algorithmic variants, with and without Hessian smoothness, clarify trade-offs in assumptions.
* Experiments show consistent gains across tasks and norms, matching the theoretical expectations.
* Clear connection between theory and experiment, with helpful discussions and well-organized comparisons to prior work.

**Weaknesses:**

* Experiments are small-scale and do not test large or modern models where Hessian-vector products may become costly.
* No wall-clock or computational cost analysis to justify practical efficiency compared to first-order baselines.
* Missing comparisons with strong modern baselines like STORM, MARS, or adaptive optimizers such as Adam.
* Theoretical analysis focuses on βₖ = 1 − αₖ, while experiments use different βₖ values without full explanation or theoretical support.
* Limited discussion of hyperparameter sensitivity and robustness, especially regarding smoothness constants and learning rates.
* Some algorithmic details and choices (norms, LMO sets, βₖ schedules) are not fully specified for reproducibility.

**Questions:**

1. How expensive are the Hessian-vector products in practice compared to standard gradient steps, in wall-clock time?
2. What strategy was used for choosing βₖ in experiments, and how stable are the results to different values?
3. Could the authors include comparisons with other O(1/K^{1/3}) methods like STORM or MARS?
4. How do these methods scale to larger models such as Transformers or large CNNs?
5. Do the observed training improvements also appear in validation or test performance?

---

> ### Author Response · Authors · 2025-11-21
>
> Thank you for your comments! Your comments are very helpful for us to improve the quality of our paper. We address your concerns below.
>
> **Computational complexity of Hessian-vector products:** The Hessian-vector product, $\nabla^2 f(x) v$ for arbitrary $x,v \in \mathbb{R}^d$, can be computed efficiently without evaluating explicit Hessians. By utilizing the automatic differentiation package in Pytorch [1], the product is evaluated via a double backward pass using the identity:
>
> $$
> \nabla^2 f(x) v = \nabla_x \langle \nabla_x f(x), v \rangle.
> $$
>
> Consequently, the computational complexity is $\mathcal{O}(d)$ where $d$ is the number of model parameters, scaling linearly. In practice, the cost of this operation is approximately equivalent to 2-3 gradient evaluations, ensuring the method remains scalable for high-dimensional models.
>
> [1] Paszke, Adam, et al. "Automatic differentiation in pytorch." (2017).
>
> **Convergence with respect to the training time:** Total training time is the product between per-iteration cost and iteration complexity. While the Hessian-vector product introduces a constant-factor overhead per iteration (roughly 2-3 times higher than the first-order baselines), second-order momentum improves the convergence rate, thus reducing the total number of iteration required to reach target accuracy. In the revised manuscript, we will add the discussion on this trade-off.
>
> **Empirical convergence against first-order baselines (STORM, MARS, and Adam):** Our LMO-based algorithms with second-order momentum achieve the convergence rate of $\mathcal{O}(1/K^{1/3})$ in the expected gradient norm. This matches existing convergence guarantees of other variance-reduced momentum algorithms, such as STORM and MARS. Given identical theoretical stepsizes, we expect our algorithms to yield comparable convergence performance.
> Regarding Adam, we acknowledge that adaptive stepsizes often accelerate empirical convergence. Our current analysis focuses on constant stepsizes. However, we agree that integrating adaptive stepsizes is the next vital step. We will expand the future work section to outline the possibility of integrating adaptive stepsizes inspired by STORM, such as $\gamma_t = \gamma/(\epsilon + \sum_t \| \nabla f(x_t;\xi_t) \|)^{1/3}$, to further enhance the performance of our proposed algorithms.
>
> **Theoretical comparisons with other O(1/K^{1/3}) methods like STORM or MARS:** From Table 1, our algorithms achieve the same $\mathcal{O}(1/K^{1/3})$ rate as STORM and MARS. However, our analysis provides two significant theoretical generalizations over these baselines. That is, our theory is established under general norms and generalized smoothness assumptions, thus ensuring that our proposed algorithms can be applied to solve problems under non-Euclidean geometries and modern learning problems like neural network training problems. In Table 1 in the revised manuscript, we explicitly compare our algorithms with STORM and MARS.
>
> **The use of $\beta_k$ values:** Inspired by the empirical benefits of $\beta_k$ that improves convergence performance of MARS over original STORM algorithms, we incorporated $\beta_k$ into our proposed algorithms. Our empirical evidence in Figure 1 supports the utility of $\beta_k$ values in our proposed algorithms. While our analysis focuses on the case when $\beta_k=1-\alpha_k$, our theory can be generalized for general $\beta_k$ values. Consequently, the resulting convergence bound with any $\beta_k$ values will exhibit one additional error term resulting from the momentum update:
>
> $$
> e_{k+1} = (1-\alpha_k)e_k  + (1-\alpha_k)W_{k+1} + \alpha_k V_{k+1} + (1-\alpha_k)(1-\beta_k/(1-\alpha_k))\nabla^2 f_{\xi_{k+1}}(x_{k+1} - x_k),
> $$
>
> where $e_k=\nabla f(x_k)-m_k$ is the momentum error term. Note that the final error term is eliminated when $\beta_k=1-\alpha_k$ that our theory considers.
>
> **Algorithmic details on norms, learning rates, and $\beta_k$ values:** We used the Euclidean norm for nonconvex logistic regression and MLP tasks, and both the Euclidean norm and the $\ell_\infty$-norm for the LSTM language modeling tasks. Moreover, while our theoretical stepsize depends on $L_1$ constants, we treat the learning rate as a tunable hyperparameter throughout our experiments. Also, we selected $\beta_k=\beta=0.5$ based on empirical tuning. We performed a grid search and observed that this value yields the fastest empirical performance. We include these details in the revised manuscript.
>
>
> > **Do the observed training improvements also appear in validation or test performance?**
>
> We focused on **Training Loss** and **Gradient Norm** because these are the direct proxies for optimization speed and convergence, which are the central claims of our theorems. While validation metrics (like perplexity) are crucial for generalization, they can be confounded by regularization and architecture; training metrics provide the purest view of the optimizer's behavior.

---

### Meta-Review · Area_Chair_byzG · 2026-01-07

**Summary:**

1. Empirical scope is too small to support claims about practical impact on modern/large models.
2. No wall-clock/runtime or compute–efficiency analysis, despite Hessian-vector products (HVPs) adding overhead.
3. Missing comparisons to strong modern baselines (STORM/MARS and/or Adam/adaptive methods), weakening empirical positioning.
4. Confusing/unclear connection to variance-reduction methods and unclear role/necessity of the MARS-style scaling factor.
5. Theory–experiment mismatch for momentum/scaling schedules (analysis assumes a specific relationship, experiments use different values) and unclear hyperparameter selection/robustness.
6. Reproducibility/details are underspecified (norm choices, LMO sets, schedules), and sensitivity to constants/learning rates is not well discussed.
7. Concerns about whether the “arbitrary norm” theory meaningfully improves geometry adaptivity (possible hidden dimension dependence in constants), and whether results are genuinely beyond prior Euclidean analyses.

**Reviewer Concerns:**

Many of the reviewer concerns have been addressed by the rebuttal, while the following concerns are still outstanding:
1. Empirical scope is too small to support the theory.
2. The reviewers requested empirical comparison between the proposed method and baselines such as STORM/MARS. The authors did not provide empirical evidence, but only argue through the theoretical guarantee about the convergence rates.
3. The Reviewer J27p's concern about implicit dimension dependence is not fully addressed by the authors' response. As pointed out by the reviewer, $\bar\rho$ can scale as $\sqrt{d}$ in the $\ell_\infty$ case, leading to dependence on $d$ that can not be ignored.

**Reviewer Scores:**

1. The Reviewer 6CUC's concerns are partially addressed by the rebuttal, and the concerns about large-scale experiments and comparison to baselines are not resolved. Their score would remain 6.
2. The Reviewer J27p's concerns are not fully addressed. In particular, the concern about the implicit dimension dependence seems critical. Thus the reviewer's score would remain 4.
3. The Reviewer FMro's concerns are partially addressed by the rebuttal, and the concerns about limited empirical scope are not resolved. Their score would remain 6.
4. The Reviewer jQAq's concerns are partially addressed by the rebuttal, and their question about empirical performance for matrix norms is not resolved. Their score would remain 8, while the confidence score is only 2.
5. The Reviewer 8Ycx's concerns are partially addressed by the rebuttal, and the concern about small-scale experiments is not resolved. Their score would remain 6, while the confidence score is only 1.

---

### Decision · Program_Chairs · 2026-01-26

Reject